# Multiclass Performance Metric Elicitation

**Gaurush Hiranandani**
Department of Computer Science
University of Illinois at Urbana-Champaign
gaurush2@illinois.edu

**Shant Boodaghians**
Department of Computer Science
University of Illinois at Urbana-Champaign
boodagh2@illinois.edu

**Ruta Mehta**
Department of Computer Science
University of Illinois at Urbana-Champaign
rutameht@illinois.edu

**Oluwasanmi Koyejo**
Department of Computer Science
University of Illinois at Urbana-Champaign
sanmi@illinois.edu

## Abstract

Metric Elicitation is a principled framework for selecting the performance metric that best reflects implicit user preferences. However, available strategies have so far been limited to binary classification. In this paper, we propose novel strategies for eliciting multiclass classification performance metrics using only relative preference feedback. We also show that the strategies are robust to both finite sample and feedback noise.

## 1 Introduction

Consider a machine learning model for cancer diagnosis and treatment support where the doctor applies a cost-sensitive predictive model to classify patients into cancer categories [23, 24]. It is clear that the chosen costs directly determine the model decisions, and thus dictate the patient outcomes. This raises an obvious question, ***how should the cost-tradeoffs be chosen so that it reflects the expert's decision-making?*** As it turns out, going from expert intuition to precise quantitative cost trade-offs is often difficult. Needless to say, this is not only true for medical applications as there are a plethora of domains where the question of *'what to measure'* poses a serious ongoing challenge [3].

To address this issue, Hiranandani et al. [7] recently formalized the problem of *Metric Elicitation (ME)*, which aims to determine the user's performance metric based on preference feedback. The motivation behind ME is that employing the performance metrics which reflect innate user tradeoffs will allow one to learn models that best capture user preferences. As humans are often inaccurate in providing absolute quality feedback [17], Hiranandani et al. [7] propose to use pairwise comparison queries, where the user (oracle) is asked to compare two classifiers and provide an indicator of relative preference. They show that in various settings, the user's innate metric can be elicited based on this preference feedback. Figure 1 (reproduced from Hiranandani et al. [7]) illustrates this framework.

Conceptually, ME is applicable to any learning setting. However, Hiranandani et al. [7] only proposed methods for eliciting binary classification performance metrics. This manuscript extends prior work by proposing ME strategies for the more complicated multiclass classification setting – thus significantly increasing the use cases for ME. Similar to the binary case, we also consider the most common families of performance metrics which are functions of the confusion matrix [15]; however, in our case, the elements of the confusion matrix summarize multiclass error statistics.

In order to perform efficient multilcass performance metric elicitation, we study novel geometric properties of the space of multiclass confusion matrices. Our analysis reveals that due to structural differences between the space of binary and multiclass confusions, we can not trivially extend the elicitation procedure used for binary to the multiclass case. Instead, we provide novel strategies for

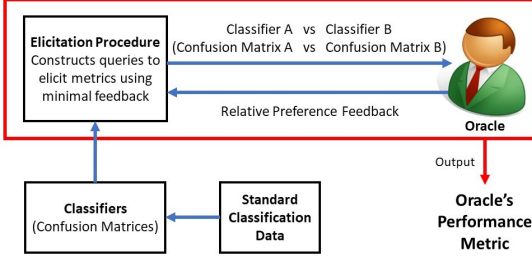

Figure 1: Metric Elicitation framework [7].

Table 1: The Bayes Optimal (BO) and Restricted-Bayes Optimal (RBO).

| Name | Definition |
|---|---|
| BO confusion $\bar{\mathbf{c}}$ over a subset $\mathcal{S} \subseteq \mathcal{C}$ | $\operatorname{argmax}_{\mathbf{c} \in \mathcal{S} \subseteq \mathcal{C}} \phi(\mathbf{c})$ |
| RBO classifier $\bar{h}_{k_1,k_2}$ | $\underset{h \in \mathcal{H}_{k_1,k_2}}{\operatorname{argmax}} \psi(\mathbf{d}(h))$ |
| RBO diagonal confusion $\bar{\mathbf{d}}_{k_1,k_2}$ | $\underset{\mathbf{d} \in \mathcal{D}_{k_1,k_2}}{\operatorname{argmax}} \psi(\mathbf{d})$ |

eliciting linear functions of the multiclass confusion matrix and extend elicitation to more complicated yet popular functional forms such as linear-fractional functions of the confusion matrix elements [14]. Specifically, the elicitation procedures involve binary-search type algorithms that are robust to both finite sample and oracle feedback noise. In addition, the proposed methods can be applied either by querying pairwise classifier preferences or pairwise confusion matrix preferences. We find that this equivalence is crucial for practical applications.

In summary, our main contributions are novel query efficient metric elicitation algorithms for multiclass classification. We study ME for linear functions of the confusion matrix and then briefly discuss extensions to more complicated functional forms such as the linear-fractional and arbitrary monotonic functions of the confusion matrix (with details in the appendix). Lastly, we show that the proposed procedures are robust to finite sample and feedback noise, thus are useful in practice.

**Notation.** Matrices and vectors are denoted by bold upper case and bold lower case letters, respectively. Let $\mathbb{R}$ and $\mathbb{Z}_+$ denote the set of reals and positive integers, respectively. For $k \in \mathbb{Z}_+$, we denote the index set $\{1, 2, \cdots, k\}$ by $[k]$. $\Delta_k$ denotes the $(k-1)$ dimensional simplex. $\|\cdot\|_1, \|\cdot\|_2$, and $\|\cdot\|_\infty$ denote the $\ell_1$-norm, $\ell_2$-norm, and $\ell_\infty$-norm, respectively. We denote the inner product of two vectors by $\langle \cdot, \cdot \rangle$. Given a matrix $\mathbf{A}$, $\textit{off-diag}(\mathbf{A})$ returns a vector of off-diagonal elements of $\mathbf{A}$ in row-major form, and $diag(\mathbf{A})$ returns a vector of diagonal elements of $\mathbf{A}$.

## 2 Preliminaries

The standard multiclass classification setting comprises $k$ classes with $X \in \mathcal{X}$ and $Y \in [k]$ representing the input and output random variables, respectively. We have access to a dataset of size $n$ denoted by $\{(\mathbf{x}, y)_i\}_{i=1}^n$, generated *iid* from a distribution $\mathbb{P}(X, Y)$. Let $\eta_i(\mathbf{x}) = \mathbb{P}(Y = i | X = \mathbf{x})$ and $\zeta_i = \mathbb{P}(Y = i)$ for $i \in [k]$ be the conditional and the unconditional probability of the $k$ classes, respectively. Let $\mathcal{H} = \{h : \mathcal{X} \to \Delta_k\}$ be the set of all classifiers. A confusion matrix for a classifier $h$ is denoted by $\mathbf{C}(h, \mathbb{P}) \in \mathbb{R}^{k \times k}$, where its elements are given by:

$$C_{ij}(h, \mathbb{P}) = \mathbb{P}(Y = i, h = j) \quad \text{for } i, j \in [k]. \tag{1}$$

Under the population law $\mathbb{P}$, it is useful to keep the following decomposition in mind:

$$\mathbb{P}(Y = i, h = i) = \zeta_i - \mathbb{P}(Y = i, h \neq i) \implies C_{ii}(h, \mathbb{P}) = \zeta_i - \sum_{j=1, j \neq i}^{k} C_{ij}(h, \mathbb{P}). \tag{2}$$

Using this decomposition, any confusion matrix is uniquely represented by its $q := (k^2 - k)$ off-diagonal elements. Hence, we will represent a confusion matrix $\mathbf{C}(h, \mathbb{P})$ by a vector $\mathbf{c}(h, \mathbb{P}) = \textit{off-diag}(\mathbf{C}(h, \mathbb{P}))$, and interchangeably refer the confusion matrix as a vector of *'off-diagonal confusions'*. The space of off-diagonal confusions is denoted by $\mathcal{C} = \{\mathbf{c}(h, \mathbb{P}) = \textit{off-diag}(\mathbf{C}(h, \mathbb{P})) : h \in \mathcal{H}\}$. For clarity, we will suppress the dependence on $\mathbb{P}$ and $h$ if it is clear from the context.

Performance of a classifier is often determined by just the misclassification and not the type of misclassification, especially when the number of classes is large. Therefore, we will also consider metrics that only depend on correct and incorrect predictions, namely $\mathbb{P}(Y = i, h = i)$ and $\mathbb{P}(Y = i, h \neq i)$. Following the decomposition in (2), such metrics require only the diagonal elements of the original confusion matrices. Given a confusion matrix $\mathbf{C}$, we will denote its diagonal by $\mathbf{d} = diag(\mathbf{C})$ and refer it as the vector of *'diagonal confusions'*. The space of diagonal confusions is represented by $\mathcal{D} = \{\mathbf{d} = diag(\mathbf{C}(h)) : h \in \mathcal{H}\}$.

Let $\phi : [0,1]^q \to \mathbb{R}$ and $\psi : [0,1]^k \to \mathbb{R}$ be the performance metrics for a classifier $h$ determined by its corresponding off-diagonal and diagonal confusion entries $\mathbf{c}(h)$ and $\mathbf{d}(h)$, respectively. Without loss of generality (wlog), we assume the metrics $\phi$ and $\psi$ are utilities so that larger values are preferred. Furthermore, the metrics are scale invariant as global scale does not affect the learning problem [15]. For this manuscript, we assume the following regularity assumption on the data distribution.

**Assumption 1.** *We assume that the functions* $g_{ij}(r) = \mathbb{P}\left[\frac{\eta_i(X)}{\eta_j(X)} \geq r\right] \forall\, i, j \in [k]$ *are continuous and strictly decreasing for* $r \in [0, \infty)$.

Intuitively, this weak assumption ensures that when the cost or reward tradeoffs for the classes change, the preferred confusion matrices for those cost or reward tradeoffs also change (and vice-versa).

## 2.1 Bayes Optimal and Restricted Bayes Optimal Confusions and Classifiers

As illustrated in Table 1, the Bayes Optimal (BO) confusion $\bar{\mathbf{c}}$ represents the optimal value of the off-diagonal confusions according to the metric $\phi$ over a subset $\mathcal{S} \subseteq \mathcal{C}$. This is analogously defined for $\psi$ and $\mathcal{D}$. The Restricted Bayes Optimal (RBO) entities are of interest for diagonal metrics $\psi$, and indicate the case where classifiers are 'restricted' to predict only classes $k_1, k_2 \in [k]$. Thus $\mathcal{H}_{k_1,k_2}$ and $\mathcal{D}_{k_1,k_2}$ denote the space of classifiers which exclusively predict either $k_1$ or $k_2$ and the associated space of diagonal confusions, respectively. Note that for such restricted classifiers $h$, $C_{ii}(h) = d_i(h)$ evaluates to zero at every index $i \neq k_1, k_2$.

## 2.2 Performance Metrics

We first discuss elicitation for the following two major types of metrics used in classification.

**Definition 1.** *Diagonal Linear Performance Metric (DLPM): We denote this family by* $\varphi_{DLPM}$. *Given* $\mathbf{a} \in \mathbb{R}^k$ *such that* $\|\mathbf{a}\|_1 = 1$ ( *wlog., due to scale invariance), the metric is defined as:* $\psi(\mathbf{d}) := \langle \mathbf{a}, \mathbf{d} \rangle$. *This is also called weighted accuracy [15] and focuses on correct classification.*

**Definition 2.** *Linear Performance Metric (LPM): We denote this family by* $\varphi_{LPM}$. *Given* $\mathbf{a} \in \mathbb{R}^q$ *such that* $\|\mathbf{a}\|_2 = 1$ *(wlog., due to scale invariance), the metric is defined as:* $\phi(\mathbf{c}) := \langle \mathbf{a}, \mathbf{c} \rangle$. *Cost-sensitive linear metrics belong to* $\varphi_{LPM}$ *[1] and focus on the types of misclassifications.*

The difference of norms in the definitions is only for simplicity of exposition and chosen to best complement the underlying metric elicitation algorithm and vice-versa. Moreover, notice that the elements of diagonal confusions ($\mathbf{d}$'s) and off-diagonal confusions ($\mathbf{c}$'s) reflect correct and incorrect classification, respectively. Thus, according to standard practice, wlog., we focus on eliciting monotonically increasing DLPMs and monotonically decreasing LPMs in their respective arguments.

## 2.3 Metric Elicitation; Problem Setup

This section describes the problem of *Metric Elicitation* and the associated *oracle query*. Our definitions follow from Hiranandani et al. [7], extended so the confusion elements and the performance metrics correspond to the multiclass classification setting. The following definitions hold analogously for the diagonal case by replacing $\phi$, $\mathbf{c}$ and $\mathcal{C}$ by $\psi$, $\mathbf{d}$, and $\mathcal{D}$, respectively.

**Definition 3** (Oracle Query). *Given two classifiers* $h, h'$ *(equivalent to off-diagonal confusions* $\mathbf{c}, \mathbf{c}'$ *respectively), a query to the Oracle (with metric* $\phi$*) is represented by:*

$$\Gamma(h, h') = \Omega(\mathbf{c}, \mathbf{c}') = \mathbb{1}[\phi(\mathbf{c}) > \phi(\mathbf{c}')] =: \mathbb{1}[\mathbf{c} \succ \mathbf{c}'], \tag{3}$$

*where* $\Gamma : \mathcal{H} \times \mathcal{H} \to \{0,1\}$ *and* $\Omega : \mathcal{C} \times \mathcal{C} \to \{0,1\}$. *The query asks whether* $h$ *is preferred to* $h'$ *(equivalent to* $\mathbf{c}$ *is preferred to* $\mathbf{c}'$*), as measured by* $\phi$.

We elicit metrics which are functions of the confusion matrix, thus comparison queries using classifiers are indistinguishable from comparison queries using confusions. Henceforth, for simplicity of notation, we denote any query as confusions based query. Next, we formally state the ME problem.

**Definition 4** (Metric Elicitation with Pairwise Queries (given $\{(\mathbf{x}, y)_i\}_{i=1}^n$)). *Suppose that the oracle's (unknown) performance metric is* $\phi$. *Using oracle queries of the form* $\Omega(\hat{\mathbf{c}}, \hat{\mathbf{c}}')$, *where* $\hat{\mathbf{c}}, \hat{\mathbf{c}}'$ *are the estimated off-diagonal confusions from samples, recover a metric* $\hat{\phi}$ *such that* $\|\phi - \hat{\phi}\| < \kappa$ *under a suitable norm* $\|\cdot\|$ *for sufficiently small error tolerance* $\kappa > 0$.

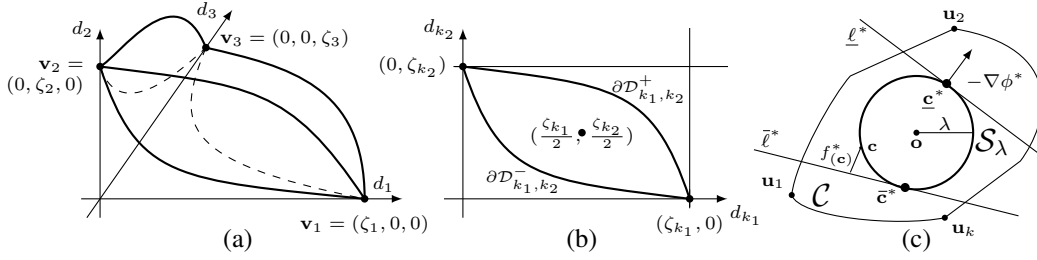

Figure 2: (a) Geometry of the space of diagonal confusions $\mathcal{D}$ for $k = 3$: a strictly convex space. Notice that each of the three axis-aligned faces are equivalent in geometry to the following figure in (b); (b) Geometry of diagonal confusions when restricted to classifiers predicting only classes $k_1$ and $k_2$ i.e. $\mathcal{D}_{k_1,k_2}$; (c) A sphere $S_\lambda$ centered at $\mathbf{o}$ with radius $\lambda$, contained in the convex space of off-diagonal confusions $\mathcal{C}$. $f^*(\mathbf{c})$ denotes the distance of $\mathbf{c}$ from the hyperplane $\bar{\ell}^*$ tangent at $\bar{\mathbf{c}}^*$.

The performance of ME is evaluated both by the fidelity of the recovered metric and the query complexity. Given the formal definitions, we can now proceed. As is standard in the decision theory literature [13, 7], we present our ME solution by first assuming access to population quantities such as the population confusions $\mathbf{c}(h, \mathbb{P})$, then examine practical implementation by considering the estimation error from finite samples e.g. with empirical confusions $\hat{\mathbf{c}}(h, \{(\mathbf{x}, y)_i\}_{i=1}^n)$.

## 3   Geometry and Parametrizations of the Query Spaces

For any query based approach, it is important to understand the structure of the query space. Thus, we first study the properties of the query spaces and then develop parametrizations required for efficient elicitation. Readers may find these properties independently useful in other applications as well.

### 3.1   Geometry of the space of diagonal confusions $\mathcal{D}$ and parametrization of its boundary

Let $\mathbf{v}_i \in \mathbb{R}^k$ for $i \in [k]$ be the vectors with $\zeta_i$ at the $i$-th index and zero everywhere else. Notice that $\mathbf{v}_i$'s are the diagonal confusions of the trivial classifiers predicting only class $i$ on the entire space $\mathcal{X}$.

**Proposition 1** (Geometry of $\mathcal{D}$ – Figure 2 (a)). *Under Assumption 1, the space of diagonal confusions $\mathcal{D}$ is strictly convex, closed, and contained in the box $[0, \zeta_1] \times \cdots \times [0, \zeta_k]$. The diagonal confusions $\mathbf{v}_i \forall i \in [k]$ are the only vertices of $\mathcal{D}$. Moreover, for any $k_1, k_2 \in [k]$, the 2-dimensional $(k_1, k_2)$ axes-aligned face of $\mathcal{D}$ is $\mathcal{D}_{k_1,k_2}$ (Figure 2 (b)), which is equivalent to the space of binary classification confusion matrices confined to classes $k_1, k_2$. In particular, $\mathcal{D}_{k_1,k_2}$ is strictly convex.*

Proposition 1 characterizes the geometry of the space of diagonal confusions $\mathcal{D}$. Figure 2(a) illustrates this geometry when $k = 3$. Interestingly, the 2-dimensional axes-aligned faces of $\mathcal{D}$ (Figure 2 (b)) have exactly the same geometry as the space of binary classification confusion matrices (compare this with Figure 2(a) of Hiranandani et al. [7]), where recall that a binary classification confusion matrix is uniquely determined by its two diagonal elements due to (2). We will exploit the set $\mathcal{D}_{k_1,k_2}$ (more specifically, its boundary) for the elicitation task. Now notice that for $\psi \in \varphi_{DLPM}$, the RBO classifier restricted to predict classes $k_1, k_2$, predicts the label (out of the two possible choices) that maximizes the expected utility conditioned on the instance. This is discussed below.

**Proposition 2.** *Let $\psi \in \varphi_{DLPM}$ be parametrized by $\mathbf{a}$ such that $\|\mathbf{a}\|_1 = 1$, and let $k_1, k_2 \in [k]$, then*

$$\bar{h}_{k_1,k_2}(\mathbf{x}) = \left\{ \begin{array}{ll} k_1, & \text{if } a_{k_1} \eta_{k_1}(\mathbf{x}) \geq a_{k_2} \eta_{k_2}(\mathbf{x}) \\ k_2, & o.w. \end{array} \right\}$$

*is the Restricted Bayes Optimal classifier (restricted to classes $k_1, k_2$) with respect to $\psi$.*

For a metric $\psi \in \varphi_{DLPM}$, Proposition 2 provides RBO classifiers in $\mathcal{H}_{k_1,k_2}$, which further gives us RBO diagonal confusions $\bar{\mathbf{d}}_{k_1,k_2}$ using (1). We know that this $\bar{\mathbf{d}}_{k_1,k_2}$ is unique, since any linear metric over a strictly convex domain ($\mathcal{D}_{k_1,k_2}$) is maximized at a unique point on the boundary [2]. So, given a DLPM, we have access to a unique point in the query space. This allows us to define and then parametrize a subset of the query space, specifically, the upper boundary of $\mathcal{D}_{k_1,k_2}$ through DLPMs.

**Definition 5.** *The upper boundary of $\mathcal{D}_{k_1,k_2}$, denoted by $\partial \mathcal{D}_{k_1,k_2}^+$, constitutes the RBO diagonal confusions confined to classes $k_1, k_2 \in [k]$ for monotonically increasing DLPMs ($a_i \geq 0 \,\forall\, i \in [k]$) such that at least one out of $a_{k_1}$ or $a_{k_2}$ is non-zero (i.e. $a_{k_1} + a_{k_2} > 0$).*

**Parameterizing the upper boundary $\partial \mathcal{D}_{k_1,k_2}^+$.** Let $m \in [0,1]$. Construct a DLPM by setting $a_{k_1} = m$, $a_{k_2} = 1 - m$, and $a_i = 0$ for $i \neq k_1, k_2$. By using Proposition 2 and (1), obtain its RBO diagonal confusions, which by definition lies on the upper boundary. Thus, varying $m$ in this process, parametrizes the upper boundary $\partial \mathcal{D}_{k_1,k_2}^+$. We denote this parametrization by $\nu(m; k_1, k_2)$, where $\nu : ([0,1]; k_1, k_2) \to \partial \mathcal{D}_{k_1,k_2}^+$.

## 3.2 Geometry of the space $\mathcal{C}$ and parametrization of the enclosed sphere

Recall that, unlike the diagonal case, we focus on eliciting LPMs monotonically decreasing in the elements of the off-diagonal confusions (Section 2.2). To this end, let $\mathbf{u}_i \in \mathcal{C}$ for $i \in [k]$ be the off-diagonal confusions achieved by trivial classifiers predicting only class $i$ on the entire space $\mathcal{X}$.

**Proposition 3** (Geometry of $\mathcal{C}$ – Figure 2 (c))**.** *The space of off-diagonal confusions $\mathcal{C}$ is convex and contained in the box $[0, \zeta_1]^{(k-1)} \times \cdots \times [0, \zeta_k]^{(k-1)}$. $\{\mathbf{u}_i\}_{i=1}^k$ belong to the set of vertices of $\mathcal{C}$. $\mathcal{C}$ always contains the point $\mathbf{o} = \frac{1}{k} \sum_{i=1}^k \mathbf{u}_i$ which corresponds to the off-diagonal confusions of the trivial classifier that randomly predicts each class with equal probability on the entire space $\mathcal{X}$.*

We find that the space of off-diagonal confusions $\mathcal{C}$ has quite different geometry than the diagonal case. For instance, $\mathcal{C}$ is not strictly convex. Nevertheless, since $\mathcal{C}$ is convex and always contains the point $\mathbf{o}$, we may make the following assumption. Please see Figure 2(c) for an illustration.

**Assumption 2.** *There exists a $q$-dimensional sphere $\mathcal{S}_\lambda \subset \mathcal{C}$ of radius $\lambda > 0$ centered at $\mathbf{o}$.*

Such a sphere always exists as long as the class-conditional distributions are not completely overlapping i.e. there is some signal for non-trivial classification. A method to obtain $\mathcal{S}_\lambda$ is discussed in Section 5. Now recall that the optimum for a linear function optimized over a sphere is given by the slope of the function scaled by the radius of the sphere. This is formalized as a trivial lemma below.

**Lemma 1.** *Let $\phi \in \varphi_{LPM}$ be parametrized by $\mathbf{a}$ such that $\|\mathbf{a}\|_2 = 1$, then the unique optimal off-diagonal confusion $\bar{\mathbf{c}}$ over the sphere $\mathcal{S}_\lambda$ is a point on the boundary of $\mathcal{S}_\lambda$ given by $\bar{\mathbf{c}} = \lambda \mathbf{a} + \mathbf{o}$.*

Given an LPM, Lemma 1 provides a unique point in the query space $\mathcal{S}_\lambda \subset \mathcal{C}$. This gives us an opportunity to characterize and then parametrize a subset of the query space through LPMs. Since we focus on eliciting monotonically decreasing LPMs, we parametrize the lower boundary of $\mathcal{S}_\lambda$.

**Definition 6.** *The lower boundary of $\mathcal{S}_\lambda$, denoted by $\partial \mathcal{S}_\lambda^-$, constitutes the set of optimal off-diagonal confusions over the sphere $\mathcal{S}_\lambda$ for LPMs with $a_i \leq 0 \,\forall\, i \in [q]$ (monotonically decreasing condition).*

**Parameterizing the lower boundary of the enclosed sphere $\partial \mathcal{S}_\lambda^-$.** We follow the standard method for parametrizing points on the surface of a sphere via angles. Let $\boldsymbol{\theta}$ be a $(q-1)$-dimensional vector of angles, where all the angles except the primary angle are in second quadrant, i.e. $\{\theta_i \in [\pi/2, \pi]\}_{i=1}^{q-2}$, and the primary angle is in the third quadrant, i.e. $\theta_{(q-1)} \in [\pi, 3\pi/2]$. Construct an LPM ($\|\mathbf{a}\|_2 = 1$) by setting $a_i = \Pi_{j=1}^{i-1} \sin \theta_j \cos \theta_i$ for $i \in [q-1]$ and $a_q = \Pi_{j=1}^{q-1} \sin \theta_j$. The choice of the quadrants ensures the monontonically decreasing condition i.e. $\{a_i \leq 0\}_{i=1}^q$. By using Lemma 1, obtain its BO off-diagonal confusions over the sphere $\mathcal{S}_\lambda$, which clearly lies on the lower boundary. Thus, varying $\boldsymbol{\theta}$ in this procedure, parametrizes the lower boundary $\partial \mathcal{S}_\lambda^-$. We denote this parametrization by $\mu(\boldsymbol{\theta})$, where $\mu : [\pi/2, \pi]^{q-2} \times [\pi, 3\pi/2] \to \partial \mathcal{S}_\lambda^-$.

# 4 Metric Elicitation

Using the outlined parametrizations $\{\nu, \mu\}$, we propose efficient binary-search type algorithms to elicit oracle's implicit performance metric. We will first discuss elicitation procedures with no *feedback* noise from the oracle. We will later show robustness to noisy feedback in Section 5.

## 4.1 DLPM Elicitation

The following lemma concerning a broader family of metrics is the route to our elicitation procedures. Since both linear and linear-fractional functions are quasiconcave, the lemma applies to both.

| **Algorithm 1: DLPM Elicitation** | **Algorithm 2: LPM Elicitation** |
|---|---|
| **Input:** $\epsilon > 0$, oracle $\Omega$, $\hat{a}_1 = 1$ | **Input:** $\epsilon > 0$, oracle $\Omega$, $\lambda$, and $\boldsymbol{\theta} = \boldsymbol{\theta}^{(1)}$ |
| **For** $i = 2, \cdots, k$ **do** | **For** $t = 1, 2, \cdots, T$ **do** |
|   **Initialize:** $m^a = 0, m^b = 1$. |   Set $\boldsymbol{\theta}^a = \boldsymbol{\theta}^c = \boldsymbol{\theta}^d = \boldsymbol{\theta}^e = \boldsymbol{\theta}^b = \boldsymbol{\theta}^{(t)}$. |
|   **While** $\left\| m^b - m^a \right\| > \epsilon$ **do** |   **if** $(t\%(q-1))$ Set $j = t\%(q-1)$; **else** $j = q-1$. |
|     • Set $m^c = \frac{3m^a + m^b}{4}$, $m^d = \frac{m^a + m^b}{2}$, and |   **if** $(j == q-1)$ **Initialize:** $\theta_j^a = \pi, \theta_j^b = 3\pi/2$. |
|     $m^e = \frac{m^a + 3m^b}{4}$. |   **else Initialize:** $\theta_j^a = \pi/2, \theta_j^b = \pi$. |
|     • Set $\overline{\mathbf{d}}_{1,i}^a = \nu(m^a; 1, i)$ (i.e. parametriza- |   **While** $\left\| \theta_j^b - \theta_j^a \right\| > \epsilon$ **do** |
|     tion of $\partial \mathcal{D}_{1,i}^+$ in Section 3.1). Similarly, set |     • Set $\theta_j^c = \frac{3\theta_j^a + \theta_j^b}{4}, \theta_j^d = \frac{\theta_j^a + \theta_j^b}{2}$, and $\theta_j^e = \frac{\theta_j^a + 3\theta_j^b}{4}$. |
|     $\overline{\mathbf{d}}_{1,i}^c, \overline{\mathbf{d}}_{1,i}^d, \overline{\mathbf{d}}_{1,i}^e, \overline{\mathbf{d}}_{1,i}^b$. |     • Set $\overline{\mathbf{c}}^a = \mu(\boldsymbol{\theta}^a)$ (i.e. parametrization of $\partial \mathcal{S}_\lambda^-$ in |
|     • Query $\Omega(\overline{\mathbf{d}}_{1,i}^c, \overline{\mathbf{d}}_{1,i}^a), \Omega(\overline{\mathbf{d}}_{1,i}^d, \overline{\mathbf{d}}_{1,i}^c),$ |     Section 3.2). Similarly, set $\overline{\mathbf{c}}^c, \overline{\mathbf{c}}^d, \overline{\mathbf{c}}^e, \overline{\mathbf{c}}^b$. |
|     $\Omega(\overline{\mathbf{d}}_{1,i}^e, \overline{\mathbf{d}}_{1,i}^d)$, and $\Omega(\overline{\mathbf{d}}_{1,i}^b, \overline{\mathbf{d}}_{1,i}^e)$. |     • Query $\Omega(\overline{\mathbf{c}}^c, \overline{\mathbf{c}}^a), \Omega(\overline{\mathbf{c}}^d, \overline{\mathbf{c}}^c), \Omega(\overline{\mathbf{c}}^e, \overline{\mathbf{c}}^d), \Omega(\overline{\mathbf{c}}^b, \overline{\mathbf{c}}^e)$ |
|     • $[m^a, m^b] \leftarrow$ *ShrinkInterval-1* (responses). |     • $[\theta_j^a, \theta_j^b] \leftarrow$ *ShrinkInterval-2* (responses). |
|   Set $m^d = \frac{m^a + m^b}{2}$. Then set $\hat{a}_i = \frac{1 - m^d}{m^d} \hat{a}_1$. |   Set $\theta_j^d = \frac{1}{2}(\theta_j^a + \theta_j^b)$ and then set $\boldsymbol{\theta}^{(t)} = \boldsymbol{\theta}^d$. |
| **Output:** $\hat{\mathbf{a}} = \left( \frac{\hat{a}_1}{\|\hat{\mathbf{a}}\|_1}, \cdots, \frac{\hat{a}_k}{\|\hat{\mathbf{a}}\|_1} \right)$. | **Output:** $\hat{a}_i = \Pi_{j=1}^{i-1} \sin \theta_j^{(T)} \cos \theta_i^{(T)} \, \forall i \in [q-1]$, |
| |     $\hat{a}_q = \Pi_{j=1}^{q-1} \sin \theta_j^{(T)}$. |

**Lemma 2.** *Let $\psi : \mathcal{D} \to \mathbb{R}$ be a quasiconcave metric which is monotone increasing in all $\{d_i\}_{i=1}^k$. For $k_1, k_2 \in [k]$, let $\rho^+ : [0,1] \to \partial \mathcal{D}_{k_1,k_2}^+$ be a continuous, bijective, parametrization of the upper boundary. Then the composition $\psi \circ \rho^+ : [0,1] \to \mathbb{R}$ is quasiconcave and thus unimodal on $[0,1]$.*

**Remark 1.** *Under Assumption 1, every supporting hyperplane of $\mathcal{D}_{k_1,k_2}$ supports a unique point on the boundary $\partial \mathcal{D}_{k_1,k_2}^+$ and vice-versa (Proposition 1); therefore, the composition $\psi \circ \rho^+$ has no flat regions. In other words, the function $\psi \circ \rho^+$ is concave.*

The proof of Lemma 2 first shows that any quasiconcave metric $\psi$ defined on the space $\mathcal{D}$ is also quasiconcave on the restricted space $\mathcal{D}_{k_1,k_2}$, and then shows the quasiconcavity and thus the unimodality (due to the one-dimensional parametrization of $\partial \mathcal{D}_{k_1,k_2}^+$) of $\psi$ on a further restricted space $\partial \mathcal{D}_{k_1,k_2}^+$. Furthermore, Remark 1 reveals that the function $\psi \circ \rho^+$ is concave, allowing us to devise the following binary-search type method for elicitation.

Suppose that the oracle's metric is $\psi^* \in \varphi_{DLPM}$ parametrized by $\mathbf{a}^*$ where $\|\mathbf{a}^*\|_1 = 1$, $\{a_i^*\}_{i=1}^k \geq 0$ (Section 2.2). Using the parametrization $\nu$, Algorithm 1 returns an estimate $\hat{\mathbf{a}}$ of $\mathbf{a}^*$. It takes two classes at a time, class 1 and class $i$. Since the metric is unimodal on $\partial \mathcal{D}_{1,i}^+$ (Lemma 2), the algorithm applies binary-search in the inner while-loop to estimate the ratio $a_i^*/a_1^*$. The *ShrinkInterval-1* subroutine shrinks the interval $[m^a, m^b]$ into half based on the oracle responses in the usual binary-search way for searching the optimum (Figure 4, Appendix A). The algorithm repeats this $(k-1)$ times to estimate the ratios $\{a_2^*/a_1^*, \ldots, a_k^*/a_1^*\}$. Finally, it outputs a normalized metric estimate $\hat{\mathbf{a}}$.

## 4.2 LPM Elicitation

We now discuss LPM elicitation, where the metrics are assumed to be monotonically decreasing in the off-diagonal confusions. Unfortunately, $\partial \mathcal{C}$ may have flat regions due to lack of strict convexity, so the algorithm for the diagonal case does not apply. Instead, we consider a query space given by the sphere $\mathcal{S}_\lambda \subset \mathcal{C}$ and propose a coordinate-wise binary-search style algorithm, which is an outcome of our novel geometric characterization and the approach in Derivative-Free Optimization (DFO) [9].

Suppose that the oracle's metric is $\phi^* \in \varphi_{LPM}$ parametrized by $\mathbf{a}^*$ where $\|\mathbf{a}^*\|_2 = 1$, $\{a_i^*\}_{i=1}^q \leq 0$ (Section 2.2). Using the parametrization $\mu(\boldsymbol{\theta})$ of $\partial \mathcal{S}_\lambda^-$ (Section 3.2), Algorithm 2 returns an estimate $\hat{\mathbf{a}}$ of $\mathbf{a}^*$. In each iteration, the algorithm updates one angle $\theta_j$ keeping other angles fixed by a binary-search procedure, where again the *ShrinkInterval-2* subroutine shrinks the interval $[\theta_j^a, \theta_j^b]$ by half based on the oracle responses (Figure 5, Appendix A). Then the algorithm cyclically updates each angle until it converges to a metric sufficiently close to the true metric. The convergence is assured because, intuitively, the algorithm via a dual interpretation minimizes a smooth, strongly convex function $f^*(\mathbf{c})$ measuring the distance of the boundary points from a hyperplane $\overline{\ell}^*$, whose slope is given by $\mathbf{a}^*$ and is tangent at the BO confusion $\overline{\mathbf{c}}^*$ (see Figure 2(c)).

Table 2: DLPM elicitation at $\epsilon = 0.01$ for synthetic data. $\#Q$ denotes the number of queries.

| Classes $k = 3$ | | | Classes $k = 4$ | | |
|---|---|---|---|---|---|
| $\psi^* = \mathbf{a}^*$ | $\hat{\psi} = \hat{\mathbf{a}}$ | #Q | $\psi^* = \mathbf{a}^*$ | $\hat{\psi} = \hat{\mathbf{a}}$ | #Q |
| (0.21, 0.59, 0.20) | (0.21, 0.60, 0.20) | 56 | (0.22, 0.13, 0.14, 0.52) | (0.22, 0.13, 0.14, 0.52) | 84 |
| (0.23, 0.15, 0.62) | (0.23, 0.15, 0.62) | 56 | (0.58, 0.17, 0.08, 0.18) | (0.58, 0.17, 0.08, 0.18) | 84 |

## 5 Guarantees

We discuss robustness under the following feedback model, which is useful in practical scenarios.

**Definition 7** (Oracle Feedback Noise: $\epsilon_\Omega \geq 0$)**.** *The oracle responses correctly as long as* $|\phi(\mathbf{c}) - \phi(\mathbf{c}')| > \epsilon_\Omega$ *(analogously* $|\psi(\mathbf{d}) - \psi(\mathbf{d}')| > \epsilon_\Omega$*). Otherwise, it may provide incorrect answers.*

In other words, the oracle may respond incorrectly if the confusions are too close as measured by the metric $\phi$ (analogously $\psi$). Next, we discuss elicitation guarantees for DLPM and LPM elicitation.

**Theorem 1.** *Given* $\epsilon, \epsilon_\Omega \geq 0$*, and a 1-Lipschitz DLPM* $\psi^*$ *parametrized by* $\mathbf{a}^*$*. Then the output* $\hat{\mathbf{a}}$ *of Algorithm 1 after* $O((k-1)\log\frac{1}{\epsilon})$ *queries to the oracle satisfies* $\|\mathbf{a}^* - \hat{\mathbf{a}}\|_\infty \leq O(\epsilon + \sqrt{\epsilon_\Omega})$*, which is equivalent to* $\|\mathbf{a}^* - \hat{\mathbf{a}}\|_2 \leq O(\sqrt{k}(\epsilon + \sqrt{\epsilon_\Omega}))$ *using standard norm bounds.*

The following theorem guarantees LPM elicitation when the sphere radius dominates the oracle noise.

**Theorem 2.** *Given* $\epsilon, \epsilon_\Omega \geq 0$*, and a 1-Lipschitz LPM* $\phi^*$ *parametrized by* $\mathbf{a}^*$*. Suppose* $\lambda \gg \epsilon_\Omega$*, then the output* $\hat{\mathbf{a}}$ *of Algorithm 2 after* $O\left(z_1 \log(z_2/(q\epsilon^2))(q-1)\log\frac{\pi}{2\epsilon}\right)$ *queries satisfies* $\|\mathbf{a}^* - \hat{\mathbf{a}}\|_2 \leq O(\sqrt{q}(\epsilon + \sqrt{\epsilon_\Omega/\lambda}))$*, where* $z_1, z_2$ *are constants independent of* $\epsilon$ *and* $q$*.*

We see that the algorithms are robust to noise, and their query complexity depends linearly in the unknown entities. The term $z_1 \log(z_2/(q\epsilon^2))$ may attribute to the number of cycles in Algorithm 2, but due to the curvature of the sphere, we observe that it is not a dominating factor in the query complexity. For instance, we find that when $\epsilon = 10^{-2}$, two cycles (i.e. $T = 2(q-1)$ in Algorithm 2) are sufficient for achieving elicitation up to the error tolerance $\sqrt{q}\epsilon$. One remaining question for LPM elicitation is to select a sufficiently large value of $\lambda$. Algorithm 3 (Appendix D) provides an offline procedure to compute a $\lambda \geq \tilde{r}/k$, where $\tilde{r}$ is the radius of the largest ball contained in the set $\mathcal{C}$.

**ME with Finite Samples:** As a final step, we consider the following questions when working with finite samples: (a) do we get the correct feedback from querying $\Omega(\hat{\mathbf{c}}, \hat{\mathbf{c}}')$ instead of querying $\Omega(\mathbf{c}, \mathbf{c}')$? (b) what is the effect of $\hat{\eta}_i$'s when used in place of true $\eta_i$'s? The answers are straightforward. Since the sample estimates of confusion matrices are consistent estimators and the metrics discussed are 1-Lipschitz with respect to the confusion matrices, with high probability, we gather correct oracle feedback as long as we have sufficient samples. Furthermore, subject to regularity assumptions, Lemma 3 of Hiranandani et al. [7] shows that the errors due to using $\hat{\eta}$ affect the (binary) confusion matrices on the boundary in a controlled manner. Since Algorithm 1 uses pairwise RBO (binary) classifiers, it inherits the error guarantees in the multiclass case. Due to limited space, we do not repeat the details here. On the other hand, since Algorithm 2 does not use the boundary, its results are agnostic to finite sample error as long as the sphere is contained within the feasible region $\mathcal{C}$.

## 6 Experiments

In this section, we empirically validate the results of theorems 1 and 2 and investigate sensitivity due to finite sample estimates.[1] For the ease of judgments, we show results for $k = 3$ and $k = 4$ classes.

### 6.1 Synthetic Data Experiments

We assume a joint distribution for $\mathcal{X} = [-1, 1]$ and $\mathcal{Y} = [k]$. This is given by the marginal distribution $f_X = \mathbb{U}[-1, 1]$ and $\eta_i(x) = \frac{1}{1+e^{p_i x}}$ for $i \in [k]$, where $\mathbb{U}[-1, 1]$ is the uniform distribution on $[-1, 1]$ and $\{p_i\}_{i=1}^k$ are the parameters controlling the degree of noise in the labels. We fix $(p_1, p_2, p_3) = (1, 3, 5)$ and $(p_1, p_2, p_3, p_4) = (1, 3, 6, 10)$ for experiments with three and four classes, respectively. To verify elicitation, we first define a true metric $\psi^*$ or $\phi^*$. This specifies the query outputs of Algorithm 1 or Algorithm 2. Then we run the algorithms to check whether or not we recover the same

Table 3: LPM elicitation at $\epsilon = 0.01$ for synthetic data. $\#Q$ denotes the number of queries.

| Classes | $\phi^* = \mathbf{a}^*$ | $\hat{\phi} = \hat{\mathbf{a}}$ | #Q |
|---|---|---|---|
| 3 | (-0.37, -0.89, -0.09, -0.23, -0.04, -0.03) | (-0.37, -0.89, -0.09, -0.23, -0.04, -0.03) | 320 |
| 3 | (-0.80, -0.55, -0.18, -0.08, -0.14, -0.05) | (-0.80, -0.55, -0.18, -0.08, -0.14, -0.05) | 320 |
| 4 | (-0.90, -0.28 -0.10, -0.31, -0.04, -0.05, -0.03, -0.04, -0.02, -0.01, -0.01, -0.01) | (-0.90, -0.28, -0.10, -0.31, -0.04, -0.05, -0.03, -0.04, -0.02, -0.01, -0.01, -0.01) | 704 |
| 4 | (-0.54, -0.10, -0.62, -0.52, -0.03, -0.07, -0.11, -0.07, -0.14, -0.03, -0.03, -0.04) | (-0.55, -0.11, -0.62, -0.51, -0.03, -0.07, -0.11, -0.07, -0.14, -0.03, -0.03, -0.04) | 704 |

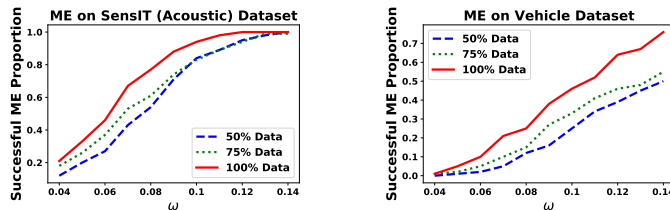

Figure 3: DLPM elicitation on real data for $\epsilon = 0.01$. For randomly chosen hundred $\mathbf{a}^*$, we show the proportion of times our estimates $\hat{\mathbf{a}}$ obtained with $4(k-1)\lceil \log(1/\epsilon) \rceil$ queries satisfy $\|\mathbf{a}^* - \hat{\mathbf{a}}\|_\infty \leq \omega$.

metric. Some results are shown in Table 2 and Table 3. Results verify that we elicit the true metrics even for small $\epsilon = 0.01$, and as predicted, this requires only $4(k-1)\lceil \log(1/\epsilon) \rceil$ and $4T\lceil \log(\pi/2\epsilon) \rceil$ queries for DLPM and LPM elicitation respectively, where $\lceil \cdot \rceil$ is the ceil function and $T = 2(q-1)$.

## 6.2 Real-World Data Experiments

Finite samples may affect the size of the sphere $S_\lambda$ in LPM elicitation, but we observe that as long as $\lambda$ is greater than $\epsilon_\Omega$ LPMs can be elicited (Appendix F.2). Thus, here we emprically validate only DLPM elicitation with finite samples. We consider two real-world datasets: (a) SensIT (Acoustic) dataset [5] (78823 instances, 3 classes), and (b) Vehicle dataset [21] (846 instances, 4 classes). From each dataset, we create two other datasets containing randomly chosen $50\%$ and $75\%$ of the datapoints. So, we have six datasets in total. For all the datasets, we standardize the features and split the dataset into two parts $\mathcal{S}_1$ and $\mathcal{S}_2$. On $\mathcal{S}_1$, we learn $\{\hat{\eta}_i(x)\}_{i=1}^k$ using a regularized softmax regression model. We use $\mathcal{S}_2$ for making predictions and computing sample confusions.

We randomly selected 100 DLPMs i.e. $\mathbf{a}^*$'s. We then used Algorithm 1 with $\epsilon = 0.01$ to recover the estimates $\hat{\mathbf{a}}$'s. In Figure 3, we show the proportion of times $\|\mathbf{a}^* - \hat{\mathbf{a}}\|_\infty \leq \omega$ for different values of $\omega$. We see improved elicitation as we increase the number of datapoints in both the datasets, suggesting that ME improves with larger datasets. In particular, for the full SensIT (Acoustic) dataset, we elicit all the metrics within $\omega = 0.12$. We also observe that $\omega \in [0.04, 0.08]$ is an overly tight evaluation criterion that can result in failures. This is because the elicitation routine gets stuck at the closest achievable sample confusions, which need not be optimal within the (small) search tolerance $\epsilon$.

# 7 Discussion Points and Future Work

- **Extensions.** The family of human evaluation metrics is believed to be large and now that we have discussed elicitation and guarantees for linear metrics, we can certainly aim for eliciting broader metric families.

  (a) Linear-fractional metrics e.g. F-measure [15] are common in classification problems because often one measures classification quality using proportions of predictions with respect to different classes. For eliciting linear-fractional metrics, we exploit their quasiconcave and quasiconvex nature. Intuitively, we aim to get a supporting hyperplane $\bar{\ell}^*$ at the maximizer $\bar{\mathbf{c}}^*$ and a supporting hyperplane $\underline{\ell}^*$ at the minimizer $\underline{\mathbf{c}}^*$ (see Figure 2(c)), which results in two non-linear systems of equations. Then we find a common solution to both the systems resulting in the true metric in just twice the number of queries required in the linear case. Due to limited space, we defer the details of diagonal and full linear-fractional elicitation to appendices E.1 and E.2, respectively.

(b) When the oracle's metric is just monotonically increasing in diagonal confusions without even having a restricted functional form, then Algorithm 1 can return a first order approximation at the BO diagonal confusion. Notice that even this may be of high importance to practitioners. The elicitation details are discussed in Appendix E.3.

- **Practical Convenience.** Our procedures can also be applied by posing pairwise classifier comparisons directly. One way is to use A/B testing [22] where the user population acts an oracle. Another way is to use comparisons from a single expert, perhaps combined with interpretable machine learning techniques [19, 4]. We suggest the approach proposed by Narasimhan [14] for estimating the classifier associated with a given confusion matrix.
- **Advantage of Algorithm 1.** When there is a reason to restrict the metric search to DLPM e.g. due to prior knowledge, then Algorithm 1 is preferred for its lower query complexity.
- **Future Work.** We conjecture that our query complexity bounds are tight; however, we leave this detail for the future. We also plan to extend our procedures for the oracles that are only probably correct. This can be done easily by applying majority voting over repeated queries [11].

## 8 Related Work

The closest line of work to ours is Hiranandani et al. [7], who proposed the problem of ME but solved it only for a simpler setting of binary classification. As we move to multiclass performance ME, we find that the form of metrics and the complexity of the query space increases. This results in stark differences in the elicitation algorithms. Algorithm 1, which is closest to the binary approach, only works for Restricted Bayes Optimal classifiers, and Algorithm 2 requires a coordinate-wise binary-search approach. As a result, novel methods are also required to provide query complexity guarantees. The LPM elicitation problem can be posed as a Derivative-Free Optimization [9] to a certain extent, but only after exploiting the geometry as we have. In addition, passively learning linear functions using pairwise comparisons has been studied before [6, 10, 16], but these approaches fail to control sample (i.e. query) complexity and end up utilizing more queries than the active approaches [20, 8, 12]. Papers which actively control the query samples for linear elicitation, e.g. [18], exploit the query space like us in order to achieve lower query complexity. However, unlike us, [18] does not provide theoretical bounds and is also applied to a different query space.

## 9 Conclusion

We study the space of multiclass confusions and propose robust, efficient algorithms to elicit diagonal-linear and linear performance metrics using preference feedback. We extend elicitation to other families e.g. linear-fractional metrics, thus covering a wide range of metrics encountered in practice.

### Acknowledgments

Gaurush Hiranandani and Oluwasanmi Koyejo thank Microsoft Azure for providing computing credits. Shant Boodaghians and Ruta Mehta acknowledge the support of NSF via CCF 1750436.

## Footnotes

[1]A subset of results is shown here. Refer Appendix F for more results.

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
