[Supplementary Material]

# Appendices

Let $f_X$ be the marginal distribution for $\mathcal{X}$.

For the purpose of clarity in the appendices, let us replace the notation of the parametrization $\nu(m; k_1, k_2)$ of the upper boundary $\partial \mathcal{D}^+_{k_1, k_2}$ by $\nu^+(m; k_1, k_2)$. This is useful to disambiguate with the parametrization $\nu^-(m; k_1, k_2)$ of the lower boundary $\partial \mathcal{D}^-_{k_1, k_2}$, which is useful in linear-fractional elicitation.

## A   ShrinkInterval-1 and ShrinkInterval-2 Subroutines

**Subroutine *ShrinkInterval-1***
**Input:** Oracle responses for $\Omega(\overline{\mathbf{d}}^c_{1,i}, \overline{\mathbf{d}}^a_{1,i})$,
$\quad \Omega(\overline{\mathbf{d}}^d_{1,i}, \overline{\mathbf{d}}^c_{1,i}), \Omega(\overline{\mathbf{d}}^e_{1,i}, \overline{\mathbf{d}}^d_{1,i})$, and $\Omega(\overline{\mathbf{d}}^b_{1,i}, \overline{\mathbf{d}}^e_{1,i})$.
**If** $(\overline{\mathbf{d}}^a_{1,i} \succ \overline{\mathbf{d}}^c_{1,i})$ Set $m^b = m^d$.
**elseif** $(\overline{\mathbf{d}}^a_{1,i} \prec \overline{\mathbf{d}}^c_{1,i} \succ \overline{\mathbf{d}}^d_{1,i})$ Set $m^b = m^d$.
**elseif** $(\overline{\mathbf{d}}^c_{1,i} \prec \overline{\mathbf{d}}^d_{1,i} \succ \overline{\mathbf{d}}^e_{1,i})$ Set $m^a = m^c, m^b = m^e$.
**elseif** $(\overline{\mathbf{d}}^d_{1,i} \prec \overline{\mathbf{d}}^e_{1,i} \succ \overline{\mathbf{d}}^b_{1,i})$ Set $m^a = m^d$.
**else** Set $m^a = m^d$.
**Output:** $[m^a, m^b]$.

Figure 4: (Left): Formal description of the subroutine *ShrinkInterval-1*. (Right): Visual intuition of the subroutine *ShrinkInterval-1*; in search of the maximizer of a quasiconcave metric $\psi$, the subroutine shrinks the current interval to half based on oracle responses to the four queries.

**Subroutine *ShrinkInterval-2***
**Input:** Oracle responses for $\Omega(\overline{\mathbf{c}}^c, \overline{\mathbf{c}}^a), \Omega(\overline{\mathbf{c}}^d, \overline{\mathbf{c}}^c)$,
$\quad \Omega(\overline{\mathbf{c}}^e, \overline{\mathbf{c}}^d), \Omega(\overline{\mathbf{c}}^b, \overline{\mathbf{c}}^e), j \in [q]$.
**If** $(\overline{\mathbf{c}}^a \succ \overline{\mathbf{c}}^c)$ Set $\theta^b_j = \theta^d_j$.
**elseif** $(\overline{\mathbf{c}}^a \prec \overline{\mathbf{c}}^c \succ \overline{\mathbf{c}}^d)$ Set $\theta^b_j = \theta^d_j$.
**elseif** $(\overline{\mathbf{c}}^c \prec \overline{\mathbf{c}}^d \succ \overline{\mathbf{c}}^e)$ Set $\theta^a_j = \theta^c_j, \theta^b_j = \theta^e_j$.
**elseif** $(\overline{\mathbf{c}}^d \prec \overline{\mathbf{c}}^e \succ \overline{\mathbf{c}}^b)$ Set $\theta^a_j = \theta^d_j$.
**else** Set $\theta^a_j = \theta^d_j$.
**Output:** $[\theta^a_j, \theta^b_j]$.

Figure 5: Formal description of the subroutine *ShrinkInterval-2*. *ShrinkInterval-2* is same as *ShrinkInterval-1* except that it applies to the parameter $\theta_j$ and works with responses to off-diagonal confusions based queries.

Notice that both *ShrinkInterval* sub-routines work with responses to four queries, and based on the responses divides the interval into two. Since the metric dealt in Algorithm 1 is concave and unimodal (see Lemma 2 and Remark 1), four queries are required to shrink the interval into by half in every iteration. Since we use the enclosed sphere for LPM elicitation, we can shrink the interval into half based on just two queries in Algorithm 2, i.e. by querying $\Omega(\overline{\mathbf{c}}^d, \overline{\mathbf{c}}^c)$ and $\Omega(\overline{\mathbf{c}}^e, \overline{\mathbf{c}}^d)$, due to strong convexity of the sphere (see proof of Theorem 2). However, we show use of four queries in Algorithm 2 just to make the algorithms consistent for the readers to understand.

## B   Proofs of Section 3 and Some Extended Definitions

*Proof of Proposition 1.* The following are the properties of $\mathcal{D}$.

- *Convex*: Let us take two classifiers $h_1, h_2 \in \mathcal{H}$ which achieve the diagonal confusions $\mathbf{d}(h_1), \mathbf{d}(h_2) \in \mathcal{D}$. We need to check whether there exists a classifier, which achieves the off-diagonal confusion $\lambda \mathbf{d}(h_1) + (1 - \lambda) \mathbf{d}(h_2)$. Consider a classifier $h$, which with probability $\lambda$ predicts what classifier $h_1$ predicts and with probability $1 - \lambda$ predicts what classifier $h_2$ predicts.

Then the first component

$$d_1(h) = \mathbb{P}(Y = 1, h = 1)$$
$$= \mathbb{P}(Y = 1, h = h_1 | h = h_1)\mathbb{P}(h = h_1) + \mathbb{P}(Y = 1, h = h_2 | h = h_2)\mathbb{P}(h = h_2)$$
$$= \lambda d_1(h_1) + (1 - \lambda)d_1(h_2).$$

Similarly, this hold true for $d_i(h)$ for $i \in [k]$. Hence, $C$ is convex.

- *Bounded:* Since $D_i = P[Y = i, h = i] \leq \zeta_i$ for all $i \in [K]$, $\mathcal{D} \subseteq [0, \zeta_1] \times \cdots \times [0, \zeta_k]$.

- *Strictly convex and closed:* Since $\mathcal{C}$ is convex, its boundary is intersection of half spaces. Furthermore, any linear functional is maximized at the boundary of a convex set [2]. Suppose we are given a diagonal linear functional (DLPM) $\mathbf{a}$. The BO classifier $h^{\mathbf{a}}$ for that function is given by Proposition 4 (whose proof is discussed later). Let the value achieved by the corresponding $BO$ diagonal confusion $\bar{d}$ is $\alpha$. That is,

$$\alpha = \sum_{i=1}^{k} a_i d_i = \sum_{i=1}^{k} \int_{\mathcal{X}} a_i \eta_i(\mathbf{x}) \mathbb{1}[h^{\mathbf{a}}(\mathbf{x}) = i | X = \mathbf{x}] df_X.$$

  Now, if we want to construct another classifier which achieves the same value $\alpha$, there has to be some weight shift from one class to another class without changing the maximum value $\alpha$. However, note that $\mathbb{P}[a_i \eta_i(X) = a_j \eta_j(X)] = 0$ for all $i, j \in [k]$ due to Assumption 1. Hence, there is a unique maximizer of this linear functional on the boundary. Therefore, the space is strictly convex. One characterization of the boundary of the space $\partial \mathcal{D}$ can be given by BO diagonal-confusions corresponding to any linear functional $\mathbf{a}$. These diagonal confusions are achieved by the corresponding BO classifiers. Therefore, these diagonal confusions are always achievable, and the space is closed as well.

- $\mathbf{v}_i$ *are always achieved:* It is easy to see that any trivial classifier which predicts only class $i \in [k]$, will achieve the diagonal confusion defined by $\mathbf{v}_i$.

- $\mathbf{v}_i$ *are the only vertices:* Certainly, a vertex exists if (and only if) some point is supported by more than $k$ tangent hyperplanes in $k$ dimensional space. This means that the vertex is optimal for more than $k$ linear metric (linear functional). Clearly, all the metrics with slope $\mathbf{a}$ such that $a_i > a_j > 0$ and $a_l = 0 \ \forall \ l \in [k], l \neq i, j$ support $\mathbf{v}_i$. So, there are at least $k$ supporting hyperplanes at these points, which make them the vertices. Now, we show that these are the only vertices.

  Suppose there is a point other than $\mathbf{v}_i$'s which is supported by two hyperplanes given by the slopes $\mathbf{a}^1$ and $\mathbf{a}^2$. From Proposition 4 (discussed later), we can get Bayes optimal classifiers $h^{\mathbf{a}^1}$ and $h^{\mathbf{a}^2}$, which achieve the same diagonal confusions. This means that

$$\int_{\mathbf{x}: \frac{\eta_1(\mathbf{x})}{\eta_j(\mathbf{x})} \geq t_j, j \in \{2, \cdots, K\}} \eta_1(\mathbf{x}) df_X = \int_{\mathbf{x}: \frac{\eta_1(\mathbf{x})}{\eta_j(\mathbf{x})} \geq t'_j, j \in \{2, \cdots, K\}} \eta_1(\mathbf{x}) df_X, \tag{4}$$

  i.e., the first component $d_1$ should be equal for the two classifiers, where $t_j, t'_j$'s are dependent on $\mathbf{a}^1$ and $\mathbf{a}^2$. Since, these classifiers are different at least for one $j$, $t_j \neq t'_j$. This will mean that there are multiple values of $\frac{\eta_1(\mathbf{x})}{\eta_j(\mathbf{x})}$ which are not attained. This contradict with our Assumption 1 that $g_{1j}$ is strictly decreasing. By strict convexity, there are no supporting hyperplane tangent at multiple points. Hence, $\mathbf{v}_i$ are the only vertices of the set $\mathcal{D}$.

Since we take classifiers which predict only classes $k_1$ and $k_2$, the values of any diagonal confusion $\mathbf{d} \in \mathcal{D}_{k_1, k_2}$ evaluate to zero at indices except $k_1, k_2$. Therefore, the properties of the space $\mathcal{D}_{k_1, k_2}$ can be proved on similar lines to Proposition 2 of Hiranandani et al. [7]. $\square$

*Proof of Proposition 3.* The following are the properties of the space $\mathcal{C}$.

- *Convex* The space is convex follows from first point of Proposition 1.
- *Bounded:* $C_{ij} = \mathbb{P}[Y = i, h = j] \leq \mathbb{P}[Y = i] = \zeta_i$ for $i, j \in [k]$. When confusion matrices written in row major form excluding the diagonal terms, then it is easy to see that $\mathcal{C} \subseteq [0, \zeta_1]^{(k-1)} \times [0, \zeta_2]^{(k-1)} \times \cdots \times [0, \zeta_k]^{(k-1)}$.

Table 4: Bayes Optimal (BO), Inverse Bayes Optimal (IBO), Restricted Bayes Optimal (RBO), and Restricted Inverse Bayes Optimal (RIBO) entities.

| Name | Definition | Name | Definition |
|---|---|---|---|
| BO classifier $\bar{h}$ | $\mathrm{argmax}_{h\in\mathcal{H}}\,\phi(\mathbf{c}(h))$ | RBO classifier $\bar{h}_{k_1,k_2}$ | $\mathrm{argmax}_{h\in\mathcal{H}_{k_1,k_2}}\,\psi(\mathbf{d}(h))$ |
| BO utility $\bar{\tau}$ over a subset $\mathcal{S}\subseteq\mathcal{C}$ | $\max_{\mathbf{c}\in\mathcal{S}\subseteq\mathcal{C}}\,\phi(\mathbf{c})$ | RBO utility $\bar{\tau}_{k_1,k_2}$ | $\max_{\mathbf{d}\in\mathcal{D}_{k_1,k_2}}\,\psi(\mathbf{d})$ |
| BO confusion $\bar{\mathbf{c}}$ over a subset $\mathcal{S}\subseteq\mathcal{C}$ | $\underset{\mathbf{c}\in\mathcal{S}\subseteq\mathcal{C}}{\mathrm{argmax}}\,\phi(\mathbf{c})$ | RBO confusion $\bar{\mathbf{d}}_{k_1,k_2}$ | $\underset{\mathbf{d}\in\mathcal{D}_{k_1,k_2}}{\mathrm{argmax}}\,\psi(\mathbf{d})$ |
| IBO classifier $\underline{h}$ | $\mathrm{argmin}_{h\in\mathcal{H}}\,\phi(\mathbf{c}(h))$ | RIBO classifier $\underline{h}_{k_1,k_2}$ | $\mathrm{argmin}_{h\in\mathcal{H}_{k_1,k_2}}\,\psi(\mathbf{d}(h))$ |
| IBO utility $\underline{\tau}$ over a subset $\mathcal{S}\subseteq\mathcal{C}$ | $\min_{\mathbf{c}\in\mathcal{S}\subseteq\mathcal{C}}\,\phi(\mathbf{c})$ | RIBO utility $\underline{\tau}_{k_1,k_2}$ | $\min_{\mathbf{d}\in\mathcal{D}_{k_1,k_2}}\,\psi(\mathbf{d})$ |
| IBO confusion $\underline{\mathbf{c}}$ over a subset $\mathcal{S}\subseteq\mathcal{C}$ | $\underset{\mathbf{c}\in\mathcal{S}\subseteq\mathcal{C}}{\mathrm{argmin}}\,\phi(\mathbf{c})$ | RIBO confusion $\underline{\mathbf{d}}_{k_1,k_2}$ | $\underset{\mathbf{d}\in\mathcal{D}_{k_1,k_2}}{\mathrm{argmin}}\,\psi(\mathbf{d})$ |

- $\mathbf{u}_i$'s and $\mathbf{o}$ are always achieved: The classifier which always predicts class $i$, will achieve the confusion matrix $\mathbf{u}_i$. Thus, $\mathbf{u}_i \in \mathcal{C}\,\forall\,i \in [q]$. Furthermore, a classifier which predicts similar to one of the trivial classifiers with probability $1/k$ will achieve the confusions $\mathbf{o}$ (the centroid).
- $\mathbf{u}_i$'s are vertices: Any supporting hyperplane with slope $a_{1i} < a_{1j} < 0$ and $a_{1l} = 0$ for $l \in [k], l \neq i,j$ will be supported by $\mathbf{u}_1$ (corresponding to BO classifier which predict class 1). Thus, $\mathbf{u}_1$ is supported by at least $q$ hyperplanes. Thus, it becomes a vertex of the convex set. Similar is the case with other $\mathbf{u}_i$'s.

$\square$

In addition to the entities defined in Table 1, we define some more entities such as the Inverse Bayes Optimal (IBO) and Restricted Inverse Bayes Optimal (RIBO) classifiers, diagonal confusions, utility in Table 4. The six definitions on the left can be analogously described diagonal metrics and diagonal confusions. The six definitions on the right are of interest for the diagonal case. These are useful in the elicitation of linear-fractional metrics and extend certain results provided in the main paper.

Proposition 2 can be considered as a corollary of the following more general Proposition.

**Proposition 4.** *Let $\psi \in \varphi_{DLPM}$, parametrized by $\mathbf{a}$, then*

$$\bar{h}(\mathbf{x}) = \underset{i\in[k]}{\mathrm{argmax}}\,a_i\eta_i(\mathbf{x}), \quad and \quad \underline{h}(\mathbf{x}) = \underset{i\in[k]}{\mathrm{argmin}}\,a_i\eta_i(\mathbf{x}) \tag{5}$$

*are the BO and IBO classifiers w.r.t $\psi$, respectively.*

*Proof.* Let

$$\psi = \sum_i a_i d_i = \sum_i \int_{\mathcal{X}} a_i\eta_i(\mathbf{x})\mathbb{1}[h(\mathbf{x}) = i].$$

From this mathematical form, it is easy to see that the metric achieves its maximum when a class that maximizes the expected utility conditioned on the instance is predicted. That is, the metric achieves its maximum when a classifier deterministically predicts class $i$ when $i = \mathrm{argmax}_{j\in[k]}\,a_j\eta_j(x)$. This is the form of the classifier written in the proposition. Similarly, this metric is minimized when when a classifier minimizes the expected utility conditioned on the instance, by predicting class $i = \mathrm{argmin}_{j\in[k]}\,a_j\eta_j(x)$. $\square$

*Proof of Proposition 2.* Recall that classifiers which predict only class $k_1$ and $k_2$ will achieve diagonal confusions, which have zeros at every other index except $k_1, k_2$. Therefore,

$$\psi = \sum_i a_i d_i = a_{k_1}d_{k_1} + a_{k_2}d_{k_2}$$

$$= \int_{\mathcal{X}} a_{k_1}\eta_{k_1}(x)\mathbb{1}[h(x) = k_1] + \int_{\mathcal{X}} a_{k_2}\eta_{k_2}(x)\mathbb{1}[h(x) = k_2].$$

Again, using the idea used in the previous proof, the metric achieves its maximum when a class that maximizes the expected utility conditioned on the instance is predicted. Therefore,

$$\bar{h}_{k_1,k_2}(x) = \left\{ \begin{array}{ll} k_1, & \text{if } a_{k_1}\eta_{k_1}(\mathbf{x}) \geq a_{k_2}\eta_{k_2}(\mathbf{x}) \\ k_2, & o.w. \end{array} \right\}$$

is the RBO classifier (restricted to classes $k_1, k_2$) with respect to $\psi$. Furthermore, the RIBO classifier is given by $\underline{h}_{k_1,k_2}(\mathbf{x}) = k_2\mathbb{1}[\bar{h}_{k1,k_2}(\mathbf{x}) = k_1] + k_1\mathbb{1}[\bar{h}_{k_1,k_2}(\mathbf{x}) = k_2]$. RIBO classifier does exactly the opposite of RBO, i.e., it predicts class $k_1$, wherever RBO predicts class $k_2$ on the instance space $\mathcal{X}$ and vice-versa. □

*Proof of Lemma 1.* Suppose the origin is at $\mathbf{o}$ and the constrained set is the sphere $\mathcal{S}_\lambda$ with radius $\lambda$ centered at $\mathbf{o}$. We want to maximize $\langle \mathbf{a}, \mathbf{c} \rangle$ such that $\mathbf{c} \in \mathcal{S}_\lambda$. Since a linear metric over a convex set is maximized at the boundary [2], it is easy to see that $c_i = \lambda a_i$ will maximize this metric. Moving the reference point to the original origin i.e. $\mathbf{0}^q$ gives us the required answer. □

For linear-fractional elicitation, we need to parametrize the lower boundary $\partial \mathcal{D}^-_{k_1,k_2}$ and upper boundary of the sphere $\partial \mathcal{S}^+_\lambda$ as well. These parametrizations are defined below.

**Definition 8.** *The RBO diagonal confusions for DLPMs parametrized by $\mathbf{a}$ with $a_{k_1}, a_{k_2} < 0$ form the lower boundary of $\mathcal{D}_{k_1,k_2}$, denoted by $\partial \mathcal{D}^-_{k_1,k_2}$.*

**Parametrization of $\partial \mathcal{D}^-_{k_1,k_2}$.** We denote this parametrization by a function $\nu^-(m; k_1, k_2)$. Take a parameter $-1 \leq m \leq 0$. Create a DLPM $\psi$ by setting $a_{k_1} = m$, $a_{k_2} = -1 - m$, and $a_i = 0$ for $i \neq k_1, k_2 \in [k]$. RBO diagonal confusions of such DLPMs lie on the lower boundary $\partial \mathcal{D}^-_{k_1,k_2}$. As we vary $m$, we move on the lower boundary $\partial \mathcal{D}^-_{k_1,k_2}$.

**Definition 9.** *The optimal off-diagonal confusions over the sphere $S_\lambda$ for LPMs parametrized by $\mathbf{a}$ with $a_i \geq 0 \ \forall i \in [k]$ form the upper boundary of $S_\lambda$, denoted by $\partial S^+_\lambda$.*

**Parametrization of $\partial S^+_\lambda$.** The parametrization of the upper boundary $\partial S^+_\lambda$ is same as that of the lower boundary $\partial S^-_\lambda$ (Section 3.2) except that now all the angles are in the first quadrant i.e. $\{\theta_i \in [0, \pi/2]\}_{i=1}^{q-1}$, so to satisfy the condition $a_i \geq 0 \ \forall i \in [k]$.

# C   Proofs of Section 4

We write Lemma 2 in the following more general form.

**Lemma 3.** *Let $\psi : \mathcal{D} \to \mathbb{R} \, (\xi : \mathcal{D} \to \mathbb{R})$ be a quasiconcave (quasiconvex) function, which is monotone increasing in all $\{d_i\}_{i=1}^k$. For $k_1, k_2 \in [k]$, let $\rho^+ : [0,1] \to \partial \mathcal{D}^+_{k_1,k_2} \, (\rho^- : [0,1] \to \partial \mathcal{D}^-_{k_1,k_2})$ be a continuous, bijective, parametrization of the upper (lower) boundary. Then the composition $\psi \circ \rho^+ : [0,1] \to \mathbb{R} \, (\xi \circ \rho^- : [0,1] \to \mathbb{R})$ is quasiconcave (quasiconvex) and thus unimodal on the interval $[0,1]$.*

*Proof.* A function is quasiconcave iff super-level sets are convex. We already know from Proposition 1 $\mathcal{D}_{k_1,k_2}$ is convex. Moreover, any vector of diagonal confusions has zeros at every index except at indices $k_1, k_2$. Let $\psi : \mathcal{D} \to \mathbb{R}$ be a quasiconcave metric, which implies that its super-level sets $\mathcal{L}^\mathcal{D}_r(\psi) = \{\mathbf{d} \in \mathcal{D} \ : \ \psi(\mathbf{d}) \geq r\}$ are convex. Now, consider the super-level sets of $\psi$ restricted to the diagonal confusions in $\mathcal{D}_{k_1,k_2}$ i.e. $\mathcal{L}^{\mathcal{D}_{k_1,k_2}}_r(\psi) = \{\mathbf{d} \in \mathcal{D}_{k_1,k_2} \ : \ \psi(\mathbf{d}) \geq r\}$. Take any $\mathbf{d}^1, \mathbf{d}^2 \in \mathcal{L}^{\mathcal{D}_{k_1,k_2}}_r(\psi)$. Since $\mathbf{d}^1, \mathbf{d}^2 \in \mathcal{D}$ as well, they belong to the set $\mathcal{L}^\mathcal{D}_r(\psi)$, which is convex. Hence, for $t \in [0,1]$, $t\mathbf{d}^1 + (1-t)\mathbf{d}^2 \in \mathcal{L}^\mathcal{D}_r(\psi)$, which implies that $\psi(t\mathbf{d}^1 + (1-t)\mathbf{d}^2) \geq r$. Furthermore, $t\mathbf{d}^1 + (1-t)\mathbf{d}^2 \in \mathcal{D}_{k_1,k_2}$, because $\mathcal{D}_{k_1,k_2}$ is convex. By the above two arguments, we have that $t\mathbf{d}^1 + (1-t)\mathbf{d}^2 \in \mathcal{L}^{\mathcal{D}_{k_1,k_2}}_r(\psi)$. This implies that $\mathcal{L}^{\mathcal{D}_{k_1,k_2}}_r(\psi)$ is convex, and hence $\psi$ restricted to $\mathcal{D}_{k_1,k_2}$ is quasiconcave. The proof analogously follows for quasiconvex metric $\xi$.

Now, it remains to show that $\psi \circ \rho^+ : [0,1] \to \mathbb{R} \, (\psi \circ \rho^- : [0,1] \to \mathbb{R})$ is quasiconcave (quasiconvex). This can be proved by readily extending the proof of Lemma 1 of Hiranandani et al. [7] to the diagonal multiclass case. For the sake of completeness, we also provide the proof here.

We will prove the result for $\psi \circ \rho^+$ on $\partial \mathcal{D}^+_{k_1,k_2}$, and the argument for $\xi \circ \rho^-$ on $\partial \mathcal{D}^-_{k_1,k_2}$ is essentially the same. For simplicity, we drop the $+$ symbols in the notation. It is given that $\psi$ is quasiconcave. Let $S$ be some superlevel set of $\psi$. We first want to show that for any $r < s < t$, if $\rho(r) \in S$ and $\rho(t) \in S$, then $\rho(s) \in S$. Since $\rho$ is a continuous bijection, due to the geometry of $\mathcal{D}_{k_1,k_2}$, we must have — wlog — $d_{k_1}(\rho(r)) < d_{k_1}(\rho(s)) < d_{k_1}(\rho(t))$, and $d_{k_2}(\rho(r)) > d_{k_2}(\rho(s)) > d_{k_2}(\rho(t))$ (otherwise swap $r$ and $t$). Since the set $\mathcal{D}_{k_1,k_2}$ is strictly convex and the image of $\rho$ is $\partial \mathcal{D}_{k_1,k_2}$, then $\rho(s)$ must dominate (component-wise) a point in the convex combination of $\rho(r)$ and $\rho(t)$. Say that point is $z$. Since $\psi$ is monotone increasing, then $x \in S \implies y \in S$ for all $y \geq x$ component-wise. Therefore, $\psi(\rho(s)) \geq \psi(z)$. Since, $S$ is convex, $z \in S$ and, due to the argument above, $\rho(s) \in S$.

This implies that $\rho^{-1}(\partial \mathcal{D}_{k_1,k_2} \cap S)$ is an interval, and is therefore convex. Thus, the superlevel sets of $\psi \circ \rho$ are convex, so it is quasiconcave, as desired. This implies unimodaltiy as a function over the real line since a function which has more than one local maximum can not be quasiconcave (consider the super-level set for some value slightly less than the lowest of the two peaks). $\square$

## D    Proofs of Section 5

*Proof of Theorem 1.*  In Hiranandani et al. [7], it is shown that for binary classification, the inner loop of Algorithm 1 will estimate the value of $\hat{m}$ for the Bayes-optimal binary classifier corresponding to a linear metric $\mathbf{a}^* = (m^*, 1 - m^*) \in \mathbb{R}^2$, such that $|\hat{m} - m^*| < \epsilon + \sqrt{\epsilon_\Omega}$ after $O(\log \frac{1}{\epsilon})$ iterations. Now, in the multiclass case, this allows us to argue that, for any $1 \leq i < j \leq k$, we can estimate a value $m_{ij}$ such that $a_i^*/a_j^* = (1 - m_{ij})/m_{ij}$.

For the required guarantees, wlog, we assumed throughout the algorithm that $a_1 \geq a_k/2$ for all $k$. This is because, if $a_1$ does not satisy this condition, then we can always choose an index $z \in [k]$ which does satisfy this from the following procedure:

Set $z \leftarrow 1$
**for** $t = 2, 3, \cdots, k$ **do**
  Compute an estimate $\hat{m}_{tz}$ of $m_{tz}$.
  **if** $\hat{m}_{tz} < \frac{1}{2}$ **then** $z \leftarrow t$ **else** do nothing
**Output:** $z$.

Let $\varepsilon = \epsilon + \sqrt{\epsilon_\Omega}$. Now, if $\hat{m}_{tz} < \frac{1}{2}$, then $a_t^* \geq a_z^* \cdot (\frac{1}{2} - \varepsilon)/(\frac{1}{2} + \varepsilon) = \frac{1-2\varepsilon}{1+2\varepsilon}$. It can be shown that this ratio is at least $1 - 4\varepsilon$. Therefore, if $z$ is the final coordinate output, we must have that $a_z \geq (1 - 4\varepsilon)^k a_t$ for all $t$. But $(1 - 4\varepsilon)^k \approx e^{-4k\varepsilon}$, and so for $\varepsilon$ sufficiently small, we have $a_z \geq a_t/2$ for all $t$ as desired. Now that we have our assumption, we may proceed to show that the algorithm is correct. We wish to show that $\|\hat{\mathbf{a}}/|\hat{a}_z| - \mathbf{a}/|a_z|\|_\infty < O(\varepsilon)$. We have

$$\left| \frac{\hat{a}_t}{\hat{a}_z} - \frac{a_t}{a_z} \right| = \left| \frac{1 - \hat{m}_t}{\hat{m}_t} - \frac{1 - m_t}{m_t} \right| = \left| \frac{1}{\hat{m}_t} - \frac{1}{m_t} \right|$$

$$\leq \frac{1}{m_t - \varepsilon} - \frac{1}{m_t} \leq \frac{1}{m_t}\left( \frac{1}{1 - 2\varepsilon} - 1 \right) \leq 2 \cdot 2\varepsilon/(1 - 2\varepsilon) \leq 5\varepsilon$$

for $\varepsilon < 0.1$. This gives us the deisred bound. $\square$

*Proof of Theorem 2.*  Consider the geometry shown in the Figure 6 (left). This shows a function $f[-1, 1]^q \to \mathbb{R}$ which follow the trajectory of a unit semicircle (semisphere). Let $\mathbf{x}$ be a q-dimensional vector, then this function is given by:

$$f(\mathbf{x}) = 1 - \sqrt{1 - \sum_i^q x_i^2} \tag{6}$$

Intuitively, this function evaluates the distance of the points lying on the surface of the semisphere. The point $\mathbf{x}^*$ (the origin) is the unique minimizer of this function. Let us restrict the domain of this function to the points $Q = [\mathbf{x}^a, \mathbf{x}^b]$, where $\mathbf{x}^a > -1$ (component-wise) and $\mathbf{x}^b < 1$ (component-wise). Then it is easy to see that the derivative of this function:

$$\nabla f = \left( \frac{x_1}{\sqrt{1 - \sum_i^q x_i^2}}, \cdots, \frac{x_q}{\sqrt{1 - \sum_i^q x_i^2}} \right) \tag{7}$$

Figure 6: (Left): A function for the semicircle with unit radius. (Right): Visual intuition for the distance between the boundary points and tangent place at the optimal off-diagonal confusions.

is continuously differentiable on a compact domain $Q$. Thus, $\nabla f$ is Lipschitz with some Lipschitz parameter $L$ i.e.:

$$\|\nabla f(\mathbf{y}) - \nabla f(\mathbf{x})\|_2 \leq L\|y - x\|_2 \tag{8}$$

which makes the function $f$ to be $L$-smooth. In addition, we observe that:

$$f(\mathbf{x}) = 1 - \sqrt{1 - \sum_i^q x_i^2} \geq \frac{1}{2}\sum_i^q x_i^2.$$

This implies that there exists a paraboloid always below the function $f$, which by definition, makes the function $f$ a strongly convex function (say with strong convexity parameter $\tau$). Thus, this function satisfies all the requirements i.e smoothness, strong convexity, and has unique minimizer, to inherit the guarantees from Derivative Free Optimization [9]. Notice that if we apply the coordinate-wise binary search Algorithm 2, where the inner loop is run for $\log(1/\epsilon)$ queries, to minimize this function using pairwise comparison queries (i.e. the oracle responds with the point that evaluate to lesser value of $f$ out of the two), then by Theorem 5 of [9] one can guarantee that after $\frac{4L}{\tau}\log(\frac{f(\mathbf{x}^0)-f(\mathbf{x}^*)}{\epsilon^2 2qL^2/\tau})q\log(1/\epsilon)$ queries to the oracle, we can get an estimate of the minimizer $\mathbf{x}^T$ such that $f(\mathbf{x}^T) - f(\mathbf{x}^*) < 4qL^2\epsilon^2/\tau$. Notice that for this function $f(x^0) - f(x^*) = f(x^0) - 0 = f(x^0) \leq 1$.

Now, for simplicity assume $\lambda = 1$. As we discussed, LPM elicitation problem, where queries are asked on a sphere $S_\lambda$ has a dual form, where we use a $(q-1)$ dimensional bijective parametrization based on $\boldsymbol{\theta}$ to denote the points on the surface of the sphere. Notice that this parametrization is a function of $\sin$ and $\cos$ and hence it is Lipschitz as well. Due to monotonicity condition, we assume that the points lie on one orthant of the sphere. Now, suppose the true oracle's metric is denoted by $\mathbf{a}^*$, where $a_i^* = \Pi_{j=1}^{i-1}\sin\theta_j\cos\theta_i$ for $i \in [q-1]$ and $a_q^* = \Pi_{j=1}^{q-1}\sin\theta_j$. Let us denote this parametrization of LPMs by $\Upsilon$, i.e. $\mathbf{a}^* = \Upsilon(\boldsymbol{\theta}^*)$. This hyperplane is tangent to the unit sphere on a particular point whose coordinates are $\Upsilon(\boldsymbol{\theta}^*)$ itself. Since the metric is linear, by posing pairwise comparisons to the oracle, we ask which off-diagonal confusion is closer to the hyperplane. So, to reach the tangent point on the boundary of the sphere by pairwise comparisons, we are actually decreasing a distance-like function $f^*(\mathbf{c})$ shown in Figure 6 (right). This function can be represented as $f^*(\boldsymbol{\theta}) = 1 - \langle\Upsilon(\boldsymbol{\theta}^*), \Upsilon(\boldsymbol{\theta})\rangle$ where $\Upsilon(\boldsymbol{\theta}^*)$ are fixed coefficients and $\boldsymbol{\theta}$ changes in our algorithm. This is equivalent to the $f$ function discussed above. Thus using the above guarantees, after $z_1\log(z_2/(q\epsilon^2))(q-1)\log(1/\epsilon)$ queries to the oracle, where $z_1, z_2$ are constants independent on $\epsilon$ and $q$, we have:

$$\begin{aligned}
f^*(\boldsymbol{\theta}) - f^*(\boldsymbol{\theta}^*) &= f^*(\boldsymbol{\theta}) - 0 \\
&= 1 - \langle\Upsilon(\boldsymbol{\theta}^*), \Upsilon(\boldsymbol{\theta}))\rangle \\
&\leq z_3 q\epsilon^2,
\end{aligned}$$

where $z_3$ is a constant depending on curvature of the above function $f$. This implies that:

$$\begin{aligned}
\|\mathbf{a}^* - \hat{\mathbf{a}}\|_2^2 &= \|\mathbf{a}^*\|_2^2 + \|\hat{\mathbf{a}}\|_2^2 - 2\langle\mathbf{a}^*, \hat{\mathbf{a}}\rangle \\
&= 2(1 - \langle\mathbf{a}^*, \hat{\mathbf{a}}\rangle) \\
&\leq 2z_3 q\epsilon^2.
\end{aligned}$$

---

**Algorithm 3** Approximating the $\lambda$ Radius

---
1: **Input:** The center $o$ of the feasible region of classifiers.
2: **for** $j = 1, 2, \cdots, q$ **do**
3:    Let $\mathbf{e}_j$ be the standard basis vector for the $j$-th dimension.
4:    Compute the maximum $\ell_j$ such that $o + \ell_j \mathbf{e}_j$ is feasible by solving (OP1).
5: Let $CONV$ be the convex hull of $\{o \pm \ell_j \mathbf{e}_j\}_{j=1}^q$.
6: Compute the radius $r$ of the largest ball which can fit inside of $CONV$, centered at $o$.
7: **Output:** $\lambda = r$.

---

Using the inequality proved before we have that $\|\mathbf{a}^* - \hat{\mathbf{a}}\|_2 \le O(\sqrt{q}\epsilon)$. Therefore, in $O\left(T \log \frac{1}{\epsilon}\right)$, we can achieve a point $O(\sqrt{q}\epsilon)$ close to the minimizer, where the number of iterations $T \ge z_1 \log(z_2/(q\epsilon^2))(q-1)$. The term $z_1 \log(z_2/(q\epsilon^2))$ can be considered as the number of cycles, but due to the curvature of the sphere, we find that it is not a dominating factor in the query complexity. For example, when working with a sphere and $\epsilon = 10^{-2}$, two cycles (i.e. $T = 2(q-1)$ in Algorithm 2) suffices in practice. Thus, updating each $\theta_j$ twice in cycles is sufficient for obtaining the required metric.

It remains to show that, whenever the queried angle is at least $\sqrt{3\epsilon_\Omega/\lambda}$ from the optimal angle, then the oracle gives a correct response. To see this, restrict attention to the hyperplane in which the current angle is moving, say $j$, for the binary-search phase of the loop. Let $\theta_j^*$ be the optimal angle. Observe that for any $\theta_j$ such that $\lambda \cos(\theta_j - \theta_j^*) \ge \lambda - \epsilon_\Omega$, the oracle may return a false value. This is because the performance metric is a 1-Lipschitz linear map, and the optimal value on the sphere of radius $\lambda$ is $\lambda$. However, $\cos(x) \le 1 - x^2/3$, and so for $|\theta_j - \theta_j^*| \ge \sqrt{3\epsilon_\Omega/\lambda}$, we have $\lambda \cos(\theta_j - \theta_j^*) \le \lambda - \lambda(3\epsilon_\Omega/\lambda)/3 = \lambda - \epsilon_\Omega$. Therefore, so long as $|\theta_j - \theta_j^*| \ge \sqrt{3\epsilon_\Omega/\lambda}$, the oracle provides a correct answer, and the binary search proceeds in the correct direction. $\quad\square$

### D.1 Finding the Sphere $\mathcal{S}_\lambda$

Now, we discuss how a sufficiently large sphere $\mathcal{S}_\lambda$ with radius $\lambda$ may be found. Consider the following optimization problem, which is a special case of OP2 in [14]. This problem corresponds to feasiblity check problem for a given off-diagonal confusion $\mathbf{c}^0$ for small $\delta \in \mathbb{R}$.

$$\min_{\mathbf{c} \in \mathcal{C}} 0 \qquad s.t. \ \|\mathbf{c} - \mathbf{c}^0\|_2 \le \delta \tag{OP1}$$

If a solution to the above problem exists, then Algorithm 1 of [14] returns it. Basically, the approach in [14] will try to construct a classifier whose off-diagonal confusions are $\delta$-close to the given off-diagonal confusion $\mathbf{c}^0$. Hence, checking the feasibility.

Algorithm 3 computes a value of $\lambda \ge \tilde{r}/k$, where $\tilde{r}$ is the radius of the largest ball contained in the set $\mathcal{C}$. Notice that this algorithm is run offline and does not impact query complexity. Notice that the approach in [14] is consistent, thus we should get a good estimate of the sphere, provided we have sufficient samples.

**Lemma 4.** *Let $\tilde{r}$ be the radius of the largest ball centered at $o$ which fits in the feasible space of classifiers. Then Algorithm 3 returns a radius $\lambda \ge \tilde{r}/k$.*

*Proof.* Let $\ell_j$ be as computed in the algorithm, and let $\ell := \min_j \ell_j$. We must have $\ell \ge \tilde{r}$. Furthermore, the region $CONV$ contains the convex hull of $\{o \pm \ell \mathbf{e}_j\}_{j=1}^q$. But this region contains a ball of radius $\ell/\sqrt{q} = \ell/\sqrt{k^2 - k} \ge \ell/k \ge \tilde{r}/k$, and so $\lambda \ge \tilde{r}/k$. $\quad\square$

## E   Extensions

We emphasize that the goal of ME is not simply to choose between default or popularly used metrics but to elicit novel metrics which best match the oracle preferences. As the family of human evaluation metrics is believed to be large and since we already have created strategies for linear metrics, we can now certainly aim at efficient elicitation for flexible metric families. Therefore, in this section, we discuss a variety of extensions to other family of metrics.

### E.1 Diagonal Linear Fractional Performance Metric (DLFPM) Elicitation

We start by first defining the diagonal linear fractional performance metric.

**Definition 10.** *Diagonal Linear-Fractional Performance Metric (DLFPM): We denote this family by $\varphi_{DLFPM}$. Given $\mathbf{a}, \mathbf{b} \in \mathbb{R}^k$ and $b_0 \in \mathbb{R}$, the metric is defined as:*

$$\psi(\mathbf{d}) = \frac{\langle \mathbf{a}, \mathbf{d} \rangle}{\langle \mathbf{b}, \mathbf{d} \rangle + b_0}. \tag{9}$$

For any $\psi \in \varphi_{DLFPM}$, we assume that $\{a_i\}_{i=1}^k, \{b_i\}_{i=1}^k$ are not all zero simultaneously and wlog, we take $\psi(\mathbf{d}) \in [0,1]$ and monotonically increasing in all $\{d_i\}_{i=1}^k$. We also make the following regularity assumption.

**Assumption 3.** *Let $\psi \in \varphi_{DLFPM}$ parametrized by $\mathbf{a}$ and $\mathbf{b}$ (Definition 1). We assume that $a_i \geq 0$ and $a_i \geq b_i$ for all $i \in [k]$. In addition, $b_0 = \sum_i (a_i - b_i)\zeta_i$ and $\sum_i a_i = 1$.*

Equivalent to fixing $\|\mathbf{a}\|_1 = 1$, $a_i \geq 0$ for the diagonal linear case (Section 2.2), the conditions in Assumption 3 are sufficient conditions for DLFPMs to be bounded and monotonically increasing in diagonal elements of the confusion matrices. This is detailed in the following proposition.

**Proposition 5.** *The conditions in Assumption 3 are sufficient for $\psi \in \varphi_{DLFPM}$ to be bounded in $[0,1]$ and simultaneously monotonically increasing in $\{d_i\}_{i=1}^k$.*

*Proof.* We can add a large positive constant if for any $\mathbf{d} \in \mathcal{D}$, $\psi(\mathbf{d}) < 0$. The metric would remain linear fractional. So, it is sufficient to assume $\psi(\mathbf{d}) \geq 0$. Furthermore, boundedness and scale invariance of $\psi$ implies $\psi(\mathbf{d}) \in [0,1]$, without compromising the linear-fractional form. Now, we look at the sufficient conditions for monotonicity in $\{d_i\}_{i=1}^k$ and the numerator and denominator to be positive. Consider the derivative:

$$\frac{\partial \psi}{\partial d_1} = \frac{a_1}{\sum_i b_i d_i + b_0} - \frac{b_1(\sum_i a_i d_i)}{(\sum_i b_i d_i + b_0)^2} \geq 0$$

Assuming denominator is positive, we have the numerator to be positive and

$$a_1 \geq b_1 \frac{\sum_i a_i d_i}{\sum_i b_i d_i + b_0} \implies a_1 \geq b_1 \sup_{\mathbf{d} \in \mathcal{D}} \frac{\sum_i a_i d_i}{\sum_i b_i d_i + b_0} \implies a_i \geq b_i \bar{\tau}$$

The above condition is necessary. Since $\bar{\tau} \in [0,1]$, by considering all the three cases $b_i = 0, b_i > 0, b_i < 0$, the following are the sufficient conditions for monotonicity: $a_1 \geq b_1$ and $a_1 \geq 0$. Similarly, this is true for all $a_i$'s and $b_i$'s i.e. $a_i \geq b_i, a_i \geq 0 \ \forall \ i \in [k]$ for monotonically increasing DLFPMs. Furthermore, as we assumed that $\psi \in [0,1]$ i.e.

$$\frac{\sum_i a_i d_i}{\sum_i b_i d_i + b_0} \leq 1 \implies \sum_i (a_i - b_i) d_i \leq b_0$$

So, it is sufficient to take $b_0 = \sum_i (a_i - b_i)\zeta_i$ to make the metric bounded in $[0,1]$ and denominator positive. In addition, we can divide the numerator and denominator by $\sum_i a_i$ without changing the metric $\psi$. Therefore, we take $\sum_i a_i = 1$ during the elicitation task. $\square$

We consider $b_0 = \sum_i (a_i - b_i)\zeta_i$, instead of the derived condition $b_0 \geq \sum_i (a_i - b_i)\zeta_i$, which is sufficient to guarantee a unique metric bounded in $[0,1]$ for elicitation purposes (instead of one of the equivalent alternatives). Note that most existing linear-fractional metrics satisfy these conditions [7, 13, 14].

Now, suppose that the oracle's metric is $\psi^* \in \varphi_{DLFPM}$. Let $\bar{\tau}^*$ and $\underline{\tau}^*$ be the maximum and minimum value of $\psi^*$, respectively. Due to strict convexity of $\mathcal{D}$, we have a hyperplane

$$\bar{\ell}_f^* := \sum_{i=1}^k (a_i^* - \bar{\tau}^* b_i^*) d_i^* = \bar{\tau}^* b_0$$

touching the set $\mathcal{D}$ only at BO diagonal confusions $\overline{\mathbf{d}}^*$ on the upper boundary of $\mathcal{D}$, denoted by $\partial\mathcal{D}^+$. Similarly, we have a hyperplane

$$\overline{\ell}_f^* := \sum_{i=1}^{k}(a_i^* - \overline{\tau}^* b_i^*)\underline{d}_i^* = \overline{\tau}^* b_0 \tag{10}$$

which touches the set $\mathcal{D}$ only at $\underline{\mathbf{d}}^*$ (IBO diagonal confusions) on the lower boundary, denoted by $\partial\mathcal{D}^-$. See Figure 2(c) for the visual intuition, where assume that the underlying space is $\mathcal{D}$ instead of the sphere $\mathcal{S}_\lambda$.

Since LFPM is quasiconcave, Algorithm 1 returns a slope of the hyperplane, say $\overline{\mathbf{s}}$. Using that slope, we can compute the Bayes Optimal diagonal confusions $\overline{\mathbf{d}}^*$ using Proposition 4, which gives us the hyperplane $\overline{\ell}^* := \langle\overline{\mathbf{s}}, \mathbf{d}\rangle = \langle\overline{\mathbf{s}}, \overline{\mathbf{d}}^*\rangle$. This is equivalent to $\overline{\ell}_f^*$ up to a constant multiple; therefore, the true metric is the solution to the following non-linear system of equations (SoE):

$$a_i^* - \overline{\tau}^* b_i^* = \alpha\overline{s}_i \ \forall\, i \in [k], \quad \overline{\tau}^* b_0^* = \alpha\langle\overline{\mathbf{s}}, \overline{\mathbf{d}}^*\rangle \tag{11}$$

where $\alpha \geq 0$, because LHS and $\overline{s}_i$'s are non-negative. If we somehow know the true $\mathbf{a}^*$, then by using the following Proposition we can elicit the DLFPM upto a constant multiple, i.e. we can get $\hat{\psi} \approx \alpha\psi^*$, which is sufficient for the elicitation task.

**Proposition 6.** *Knowing $\mathbf{a}^*$ i.e. using $\hat{\mathbf{a}} = \mathbf{a}^*$ solves the SoEs (11) as:*

$$\hat{b}_i = (\hat{a}_i - \overline{s}_i)\frac{\Lambda_1}{\Lambda_2}, \tag{12}$$

*where $\Lambda_1 = \sum_i \hat{a}_i\zeta_i$, $\Lambda_2 = \langle\overline{\mathbf{s}}, \overline{\mathbf{d}}^*\rangle + \sum_i(\hat{a}_i - \overline{s}_i)\zeta_i$, and $\hat{b}_0$ is as defined in Assumption 3.*

*Proof.* We continue from Equation (11), where we saw that $\alpha \geq 0$. Additionally, we ignore the case when $\alpha = 0$, since this would imply a constant $\psi^*$. Next, we may divide the above equations by $\alpha > 0$ on both sides so that all the coefficients $\mathbf{a}^*$ and $\mathbf{a}^*$ are factored by $\alpha$. This does not change the metric $\psi^*$; thus, the SoE becomes:

$$a_i' - \overline{\tau}^* b_i' = \overline{s}_i \ \forall\, i \in [k], \quad \overline{\tau}^* b_0' = \langle\overline{\mathbf{s}}, \overline{\mathbf{d}}^*\rangle. \tag{13}$$

Notice that none of the conditions in Assumption 3 are changed except $\sum_i a_i = 1$. However, we may still use this condition to learn a constant $\alpha$ times the true metric, which does not harm the elicitation problem. From the last equation, we have that $\overline{\tau} = \langle\overline{\mathbf{s}}, \overline{\mathbf{d}}^*\rangle / b_0'$. Putting this into rest of the equations gives us:

$$\frac{a_i' - \overline{s}_i}{\langle\overline{\mathbf{s}}, \overline{\mathbf{d}}^*\rangle} = \frac{b_i'}{b_0'}.$$

By replacing $b_i'$ in the rest of equations further gives us the solution mentioned in the proposition. $\square$

Now the question is how do we get the true $\mathbf{a}^*$. To our rescue, we also know that a DLFPM is quasiconvex. Thus, by minimizing the metric (again by using restricted classifiers) using Algorithm 4 (described next), we can get a similar hyperplane on the lower boundary $\partial\mathcal{D}^-$. Algorithm 4 is described below.

**Algorithm 4.** *Minimizing diagonal quasiconvex metrics:* This algorithm is same as Algorithm 1 with only two changes. First, we start with $m \in [-1, 0]$, because the optimum will lie on the lower boundary $\partial\mathcal{D}^-$. Second, we check for $\mathbf{d} \prec \mathbf{d}'$ whenever Algorithm 1 checks for $\mathbf{d} \succ \mathbf{d}'$, and vice-versa. Here, we output the counterpart, i.e., slope $\underline{\mathbf{s}}$.

Once we get the slope $\underline{\mathbf{s}}$, we can obtain the inverse Bayes diagonal confusion $\underline{\mathbf{d}}^*$ using Proposition 4. This will result in a supporting hyperplane $\underline{\ell}^* := \langle\underline{\mathbf{s}}, \mathbf{d}\rangle = \langle\underline{\mathbf{s}}, \underline{\mathbf{d}}^*\rangle$. This hyperplane is tangent to the lower boundary $\partial\mathcal{D}^-$, and equivalent to $\underline{\ell}_f^*$ up to a constant multiple; thus, the true metric is also the solution of the following SoE:

$$a_i^* - \underline{\tau}^* b_i^* = \gamma\underline{s}_i \ \forall\, i \in [k], \quad \underline{\tau}^* b_0^* = \gamma\langle\underline{\mathbf{s}}, \underline{\mathbf{d}}^*\rangle$$

where $\gamma \leq 0$ since LHS is positive, but $\underline{s}_i$'s are negative. Again, we may assume $\gamma < 0$. By dividing the above equations by $-\gamma$ on both sides, all the coefficients are factored by $-\gamma$. This does not change $\psi^*$; thus, the system of equations becomes the following:

$$a_i'' - \underline{\tau}^* b_i'' = \underline{s}_i, \ \forall\, i \in [k], \quad \underline{\tau}^* b_0'' = \langle\underline{\mathbf{s}}, \underline{\mathbf{d}}^*\rangle. \tag{14}$$

---

**Algorithm 5** DLFPM: Grid Search for Best Pairwise Ratios

---
1: **Input:** $n', \delta$.
2: **for** $j = 2, \cdots, k$ **do**
3:     **Initialize:** $\sigma_{opt} = \infty, a'_j = 0$.
4:     Sample $\mathbf{d}^1, ..., \mathbf{d}^{n'}$ on $\partial \mathcal{D}_{1,j}$ (BO or IBO diagonal confusions for random $n'$ DLPMs).
5:     **for** $(a'_j = 0; a'_j \leq 1; a'_j = a'_j + \delta)$ **do**
6:         Compute $\psi', \psi''$ using Proposition 6.
7:         Compute array $r = [\frac{\psi'(\mathbf{d}^1)}{\psi''(\mathbf{d}^1)}, ..., \frac{\psi'(\mathbf{d}^{n'})}{\psi''(\mathbf{d}^{n'})}]$. Set $\sigma = \mathrm{std}(r)$.
8:         **if** $(\sigma < \sigma_{opt})$ Set $\sigma_{opt} = \sigma$ and $a'_{j,opt} = a'_j$.
9:     Set $a'_j = \frac{a'_{j,opt}}{1 - a'_{j,opt}}$.
10: $a'_1 = 1$.
11: **Output:** $\mathbf{a}' = \left( \frac{a'_1}{\|\mathbf{a}'\|_1}, \cdots, \frac{a'_k}{\|\mathbf{a}'\|_1} \right)$.

---

Now, if we know $\mathbf{a}'$ in (13), then by using Proposition 6, we may solve the system (13) and obtain a metric, say $\psi'$. System (14) can be solved analogously, provided we know $\mathbf{a}''$ in (14), to get a metric, say $\psi''$. Notice that when when we have the true ratio i.e $a_i^*/a_j^* = a'_i/a'_j = a''_i/a''_j$ for $i, j \in [k]$, then $\psi^* = \psi'/\alpha = -\psi''/\gamma$. This means that when the true ratios are known, then $\psi', \psi''$ are constant multiples of each other. So, we look for the ratios where the solution to the two systems are just pointwise constant multiple of one another. This is the same idea used in the binary case [7]. However, we have to search for the entire grid $[0, 1]^k$ instead of $[0, 1]$ as is in the binary case. This is a computationally challenging task.

Notice that we can randomly sample diagonal confusions on the boundary $\partial \mathcal{D}$. This is done by first randomly generating DLPMs and then computing their BO or IBO diagonal confusions using Proposition 4. After obtaining $\bar{\ell}^*$ and $\underline{\ell}^*$, we run the grid seacrh based Algorithm 5 to find the estimates of the true $a_i$'s. Although the grid-search based algorithm is independent of oracle queries, it is computationally efficient. It runs for $(k - 1)$ rounds, where in each round it matches the solution of the two SoE's as closely as possible on a number of samples from the boundary $\partial \mathcal{D}_{1,k}$ and figures out the ratio of $a_j/a_1$ for $j \neq 1 \in [k]$. Thanks to the property $\sum_i a_i = 1$ and access to the restricted diagonal confusions, we are saved from searching the entire grid $[0, 1]^k$ to merely $(k - 1)$ times grid-search on $[0, 1]$.

### E.2 LFPM Elicitation

We start by defining the linear-fractional performance metric in off-diagonal confusions.

**Definition 11.** *Linear-Fractional Performance Metric (LFPM): We denote this family by $\varphi_{LFPM}$. Given constants $\mathbf{a}, \mathbf{b} \in \mathbb{R}^q$ and $b_0 \in \mathbb{R}$, the metric is defined as*

$$\phi(\mathbf{c}) = \frac{\langle \mathbf{a}, \mathbf{c} \rangle}{\langle \mathbf{b}, \mathbf{c} \rangle + b_0}. \tag{15}$$

For any $\phi \in \varphi_{LFPM}$ (Definition 11), we assume that $\{a_i\}_{i=1}^q, \{b_i\}_{i=1}^q$ are not all zero simultaneously. Furthermore, wlog, we may take $\phi(\mathbf{c}) \in [-1, 0] \ \forall \ \mathbf{c} \in \mathcal{C}$ and monotonically decreasing in all $\{c_i\}_{i=1}^q$. Similar to the diagonal case, we also make the following regularity assumption.

**Assumption 4.** *Let $\phi \in \varphi_{LFPM}$ (Definition 11). We assume that $a_i \leq 0$ and $a_i \leq -b_i$ for all $i \in [q]$. In addition, $b_0 = \sum_i -(a_i + b_i)\zeta_i$, and $\sum_i a_i = -1$.*

Equivalent to fixing $\|\mathbf{a}\|_1 = 1$, $a_i \geq 0$ for the diagonal linear case (Section 2.2), the conditions in Assumption 4 are sufficient conditions for LFPMs to be bounded and monotonically decreasing in off-diagonal elements of the confusion matrices. This is detailed in the following proposition.

**Proposition 7.** *Assumption 4 is sufficient for $\phi \in \varphi_{LFPM}$ to be bounded in $[-1, 0]$ and simultaneously monotonically decreasing in $\{c_i\}_{i=1}^q$.*

*Proof.* Recall that our metric $\phi$ is monotonically decreasing in $c_i$'s. As LFPMs are transitional and scale invariant, wlog, we can assume that $\phi \in [-1, 0]$. Taking the derivative in $c_1$ gives us:

$$\frac{\partial \phi}{\partial c_1} = \frac{a_1}{\sum_i b_i a_i + b_0} - \frac{b_1(\sum_i a_i c_i)}{(\sum_i b_i c_i + b_0)^2} \leq 0$$

Assuming denominator is positive, we have the numerator to be negative and

$$a_1 \leq b_1 \frac{\sum_i a_i c_i}{\sum_i b_i c_i + b_0} \implies \leq b_1 \phi(\mathbf{c}) \implies b_1 \underline{\tau}$$

The above condition is necessary. Since $\underline{\tau} \in [-1, 0]$, by considering all the cases i.e. $b_i = 0, b_i > 0, b_i < 0$ the following are the sufficient condition for monotonicity decreasing LFPMs: $a_1 \leq -b_1$ and $a_1 \leq 0$. Similarly, this is true for $a_i \leq -b_i, a_i \leq 0 \ \forall \ i \in [q]$ for monotonically decreasing LFPMs. Furthermore, as we assumed that $\phi \in [-1, 0]$ i.e.

$$\frac{\sum_i a_i c_i}{\sum_i b_i c_i + b_0} \geq -1 \implies \sum_i -(a_i + b_i)c_i \leq b_0$$

Again, so it is sufficient to take $b_0 = \sum_i -(a_i + b_i)\zeta_i$ to make the metric bounded in $[-1, 0]$ and denominator positive. In addition, we can divide the numerator and denominator by $\sum_i |a_i|$ without changing the metric $\phi$. This gives us the condition $\sum_i a_i = -1$. □

We consider $b_0 = \sum_i -(a_i + b_i)\zeta_i$, instead of the derived condition $b_0 \geq \sum_i -(a_i + b_i)\zeta_i$, which is sufficient to guarantee a unique metric bounded in $[-1, 0]$ for elicitation purposes (instead of one of the equivalent alternatives). Note that most existing linear-fractional metrics satisfy these conditions [7, 13, 14].

Now, suppose that the oracle's metric is $\phi^* \in \varphi_{LFPM}$. Let $\bar{\tau}^*$ and $\underline{\tau}^*$ be the maximum and minimum value of $\phi^*$, respectively. Due to strict convexity of $\mathcal{S}_\lambda$, we have a hyperplane

$$\bar{\ell}_f^* := \sum_{i=1}^q (a_i^* - \bar{\tau}^* b_i^*)\bar{c}_i^* = \bar{\tau}^* b_0$$

touching the set $\mathcal{S}_\lambda$ only at BO confusions $\bar{\mathbf{c}}^*$ (over the sphere $\mathcal{S}_\lambda$) on the lower boundary $\partial \mathcal{S}_\lambda^-$. Similarly, we have a hyperplane

$$\underline{\ell}_f^* := \sum_{i=1}^q (a_i^* - \underline{\tau}^* b_i^*)\underline{c}_i^* = \underline{\tau}^* b_0 \tag{16}$$

which touches the set $\mathcal{S}_\lambda$ only at inverse Bayes Optimal confusions $\underline{\mathbf{c}}^*$ (over the sphere $\mathcal{S}_\lambda$) on the upper boundary $\partial \mathcal{S}_\lambda^+$. See Figure 2(c) for the visual intuition.

Here, we use strict convexity of $\mathcal{S}_\lambda$ and follow the same arguments as in DLFPM to get a hyerplane $\bar{\ell}^* := \langle \bar{\mathbf{s}}, \mathbf{c} \rangle = \langle \bar{\mathbf{s}}, \bar{\mathbf{c}}^* \rangle$ after using Algortihm 2. Here, $\bar{\mathbf{c}}^*$ is the optimal best (BO) off-diagonal confusion on the sphere. The only difference is that the BO confusions lie on the lower boundary $\partial \mathcal{S}_\lambda^-$ (monotonically decreasing). The SoE we get is:

$$a_i^* - \bar{\tau}^* b_i^* = \alpha \bar{s}_i \ \forall \ i \in [q], \qquad \bar{\tau}^* b_0^* = \alpha \langle \bar{\mathbf{s}}, \bar{\mathbf{c}}^* \rangle \tag{17}$$

where $\alpha \geq 0$. Similar to DLFPMs, by knowing $\mathbf{a}^*$, we can elicit the LFPM upto a constant multiple.

**Proposition 8.** *Knowing $\mathbf{a}^*$ i.e. using $\hat{\mathbf{a}} = \mathbf{a}^*$ solves the SoEs (17) as:*

$$\hat{b}_i = (\hat{a}_i - \bar{s}_i)\frac{\Lambda_1'}{\Lambda_2'}, \tag{18}$$

*where $\Lambda_1' = -\sum_i \hat{a}_i \zeta_i$, $\Lambda_2' = \langle \bar{\mathbf{s}}, \bar{\mathbf{c}}^* \rangle + \sum_i (\hat{a}_i - \bar{s}_i)\zeta_i$, and $\hat{b}_0$ is as defined in Assumption 4.*

*Proof.* We start from (17), where we saw $\alpha \geq 0$. Additionally, we ignore the case when $\alpha = 0$, since this would imply a constant $\phi^*$. Next, we may divide the above equations by $\alpha > 0$ on both sides

so that all the coefficients $a_i^*$'s and $b_i^*$'s are factored by $\alpha$. This does not change $\phi^*$; thus, the SoE becomes:

$$a_i' - \bar{\tau}^* b_i' = \bar{s}_i, \quad \forall \ i \in [q], \qquad \bar{\tau}^* b_0' = \langle \bar{\mathbf{s}}, \bar{\mathbf{c}}^* \rangle. \tag{19}$$

Notice that none of the conditions in Assumption 4 are changed except $\sum_i a_i = -1$. However, we may still use this condition to learn a constant $\alpha$ times the true metric, which does not harm the elicitation problem. Similar to DLFPMs, if we somehow know the true $a_i'$'s, we can elicit the LFPM upto a constant multiple. From the last equation, we have that $\bar{\tau} = \langle \bar{\mathbf{s}}, \bar{\mathbf{c}}^* \rangle / b_0'$. Putting this into rest of the equations gives us:

$$\frac{a_i' - \bar{s}_i}{\langle \bar{\mathbf{s}}, \bar{\mathbf{c}}^* \rangle} = \frac{b_i'}{b_0}.$$

By replacing $b_i$ in the rest of equations further gives us the solution (given $a_i$'s) mentioned in the proposition. □

Now again the question is how do we get the true $\mathbf{a}^*$. To our rescue, we also know that an LFPM is quasiconvex. Thus, by minimizing the metric using Algorithm 6 (described next), we can get a similar hyperplane $\underline{\ell}^* := \langle \underline{\mathbf{s}}, \underline{\mathbf{c}} \rangle = \langle \underline{\mathbf{s}}, \underline{\mathbf{c}}^* \rangle$ tangent to the upper boundary $\partial \mathcal{S}_\lambda^+$.

**Algorithm 6.** *Minimizing quasiconvex metrics of off-diagonal confusions:* This algorithm is same as Algorithm 2 with only two changes. First, we start with $\boldsymbol{\theta} \in [0, \pi/2]^q$, because the optimum will lie on the upper boundary $\partial \mathcal{S}_\lambda^+$. Second, we check for $\mathbf{c} \prec \mathbf{c}'$ whenever Algorithm 2 checks for $\mathbf{c} \succ \mathbf{c}'$, and vice versa. Here, we output the counterpart, i.e., slope $\underline{\mathbf{s}}$.

Thus, a similar SoE (17) whose solution looks like Proposition 8 is obtained. After obtaining $\bar{\ell}^*$ and $\underline{\ell}^*$, we run grid-search Algorithm 7 to find the estimates of the true $a_i$'s. The algebra related to LFPM elicitation is same as the DLFPM case. However, this time we need to search in $[0, 1]^{q-1}$ grid. Again, we have easy access to off-diagonal confusions on the sphere $\partial \mathcal{S}_\lambda$ corresponding to BO or IBO off-diagonal confusions for different LPMs (Lemma 1); therefore, we can use the following algorithm, which is analogous to Algorithm 5.

**Algorithm 7.** *LFPM: grid-search for pairwise ratios:* This is same as Algorithm 5 except the following two changes. First, the second line of Algorithm 5 will have a for loop running from 2 to $q - 1$. Second, in line 4, samples will be generated from the surface of the sphere $\partial \mathcal{S}_\lambda$ as discussed above, instead of $\partial \mathcal{D}_{1,k}$.

### E.3 Monotonic Metrics of diagonal confusions

Recall that the space $\mathcal{D}$ is strictly convex. Suppose that the oracle's metric is $\psi^*$, which is just monotonic increasing in $\{d_i\}_{i=1}^k$. Let $\mathbf{a}^*$ be the slope of the supporting hyperplane at the optimal diagonal confusions $\mathbf{d}^*$. Then we may use Algorithm 1 which will return a linear metric $\hat{\mathbf{a}}$ by using pairwise comparisons. Notice that, we may then compute an estimate of the BO diagonal confusions $\hat{\mathbf{d}}$ using Proposition 4 corresponding to the output $\hat{\mathbf{a}}$ of the algorithm. Since the space $\mathcal{D}$ is strictly convex, $\langle \hat{\mathbf{a}}, \mathbf{d} \rangle = \langle \hat{\mathbf{a}}, \hat{\mathbf{d}} \rangle$ becomes the estimate of the unique supporting hyperplane at $\hat{\mathbf{d}}$.

The first order approximation of $\psi^*$ at $\hat{\mathbf{d}}$ can be given by:

$$\psi^*(\mathbf{d}) = \psi^*(\hat{\mathbf{d}}) + \langle \hat{\mathbf{a}}, \mathbf{d} - \hat{\mathbf{d}} \rangle.$$

Since performance metrics are not affected by scale and additive biases, then the first order approximation given by $\langle \hat{\mathbf{a}}, \mathbf{d} \rangle$ suffices for the elicitation task. Notice that this maybe of high practical importance to practitioners, since this is an estimate of the weighted accuracy at the estimate of the optimal diagonal confusions.

## F Extended Experiments

In this section, we empirically validate the theory and investigate the sensitivity due to finite sample estimates. For the ease of judgments, we show results corresponding to $k = 3, 4$ classes.[2]

Table 5: DLPM elicitation at $\epsilon = 0.01$ for synthetic data. #$Q$ denotes the number of queries. Since the digits are rounded to two decimal places, $\|\mathbf{a}^*\|_1$ or $\|\hat{\mathbf{a}}\|_1$ might not be exactly equal to one.

| Classes $K = 3$ | | | Classes $K = 4$ | | |
|---|---|---|---|---|---|
| $\psi^* = \mathbf{a}^*$ | $\hat{\psi} = \hat{\mathbf{a}}$ | #Q | $\psi^* = \mathbf{a}^*$ | $\hat{\psi} = \hat{\mathbf{a}}$ | #Q |
| (0.21, 0.59, 0.20) | (0.21, 0.60, 0.20) | 56 | (0.13, 0.37, 0.12, 0.38) | (0.13, 0.37, 0.12, 0.38) | 84 |
| (0.44, 0.26, 0.31) | (0.44, 0.26, 0.31) | 56 | (0.21, 0.26, 0.31, 0.22) | (0.21, 0.26, 0.31, 0.22) | 84 |
| (0.46, 0.33, 0.22) | (0.46, 0.33, 0.22) | 56 | (0.23, 0.17, 0.11, 0.48) | (0.23, 0.17, 0.11, 0.48) | 84 |
| (0.23, 0.15, 0.62) | (0.23, 0.15, 0.62) | 56 | (0.25, 0.13, 0.45, 0.18) | (0.25, 0.12, 0.45, 0.18) | 84 |
| (0.31, 0.15, 0.54) | (0.3, 0.15, 0.54) | 56 | (0.22, 0.17, 0.31, 0.29) | (0.22, 0.17, 0.31, 0.29) | 84 |
| (0.29, 0.40, 0.31) | (0.29, 0.40, 0.31) | 56 | (0.38, 0.21, 0.22, 0.20) | (0.38, 0.21, 0.21, 0.20) | 84 |
| (0.35, 0.32, 0.33) | (0.35, 0.33, 0.33) | 56 | (0.22, 0.13, 0.14, 0.52) | (0.22, 0.13, 0.14, 0.52) | 84 |
| (0.33, 0.35, 0.32) | (0.33, 0.35, 0.31) | 56 | (0.58, 0.17, 0.08, 0.18) | (0.58, 0.17, 0.08, 0.18) | 84 |
| (0.45, 0.27, 0.29) | (0.45, 0.26, 0.29) | 56 | (0.32, 0.35, 0.06, 0.27) | (0.32, 0.35, 0.06, 0.27) | 84 |
| (0.44, 0.44, 0.13) | (0.45, 0.43, 0.13) | 56 | (0.05, 0.24, 0.29, 0.42) | (0.05, 0.24, 0.29, 0.42) | 84 |

Table 6: LPM elicitation at $\epsilon = 0.01$ for synthetic data. #$Q$ denotes the number of queries. Since the digits are rounded to two decimal places, $\|\mathbf{a}^*\|_2$ or $\|\hat{\mathbf{a}}\|_2$ might not be exactly equal to one.

| Classes | $\phi^* = \mathbf{a}^*$ | $\hat{\phi} = \hat{\mathbf{a}}$ | #Q |
|---|---|---|---|
| 3 | (-0.37, -0.89, -0.09, -0.23, -0.04, -0.03) | (-0.37, -0.89, -0.09, -0.23, -0.04, -0.03) | 320 |
| 3 | (-0.80, -0.55, -0.18, -0.08, -0.14, -0.05) | (-0.80, -0.55, -0.18, -0.08, -0.14, -0.05) | 320 |
| 3 | (-0.19, -0.88, -0.28, -0.10, -0.08, -0.30) | (-0.19, -0.88, -0.28, -0.10, -0.08, -0.30) | 320 |
| 3 | (-0.44, -0.55, -0.33, -0.51, -0.23, -0.28) | (-0.44, -0.55, -0.33, -0.51, -0.23, -0.28) | 320 |
| 3 | (-0.79, -0.27, -0.25, -0.21, -0.38, -0.23) | (-0.79, -0.27, -0.25, -0.21, -0.38, -0.23) | 320 |
| 4 | (-0.90, -0.28 -0.10, -0.31, -0.04, -0.05, -0.03, -0.04, -0.02, -0.01, -0.01, -0.01) | (-0.90, -0.28, -0.10, -0.31, -0.04, -0.05, -0.03, -0.04, -0.02, -0.01, -0.01, -0.01) | 704 |
| 4 | (-0.54, -0.10, -0.62, -0.52, -0.03, -0.07, -0.11, -0.07, -0.14, -0.03, -0.03, -0.04) | (-0.55, -0.11, -0.62, -0.51, -0.03, -0.07, -0.11, -0.07, -0.14, -0.03, -0.03, -0.04) | 704 |
| 4 | (-0.56, -0.07, -0.79, -0.05, -0.16, -0.16, -0.04, -0.02, -0.03, -0.00, -0.01, -0.01) | (-0.56, -0.07, -0.79, -0.05, -0.16, -0.17, -0.04, -0.02, -0.03, -0.00, -0.01, -0.01) | 704 |
| 4 | (-0.60, -0.79, -0.09, -0.01, -0.01, -0.02, -0.02, -0.01, -0.01, -0.01, -0.00, -0.00) | (-0.60, -0.79, -0.09, -0.01, -0.01, -0.02, -0.02, -0.01, -0.01, -0.01, -0.00, -0.00) | 704 |
| 4 | (-0.45, -0.38, -0.42, -0.19, -0.21, -0.63, -0.09, -0.00, -0.00, -0.00, -0.01, -0.01) | (-0.46, -0.38, -0.41, -0.19, -0.20, -0.62, -0.09, -0.00, -0.00, -0.00, -0.01, -0.01) | 704 |

## F.1 DLPM and LPM Elicitation on Simulated Data (Extended)

We show an extended set of results for the experimental setting discussed in Section 6.1. Table 5 and Table 6 show elicitation results on the simulated data for DLPMs and LPMs, respectively. We verify that our algorithms elicit the true metrics even for $\epsilon = 0.01$, and as expected, require $4(k-1)\lceil\log(1/\epsilon)\rceil$ and $4T\lceil\log(\pi/2\epsilon)\rceil$ queries for DLPM and LPM elicitation, respectively, where $\lceil\cdot\rceil$ is the ceil function and $T = 2(q-1)$.

## F.2 Effect of Sphere Size on LPM Elicitation

For real-world datasets, Algorithm 2 is agnostic to the error from $\hat{\eta}_i$'s as long as we get a sphere inside the feasible region of sufficient size. With the following experiment, we show that we incur errors in elicitation when the radius $\lambda$ is of the order of $\epsilon_\Omega$. Recall that, when we are working in a simulated setting, a good proxy for $\epsilon_\Omega$ is the practical computation error.

Here, we work with $k = 4$ classes. We took $\lambda = 2.500 \times 10^{-12}$ and performed elicitation by considering three spheres of size $1/2\lambda$, $3/4\lambda$, and $\lambda$. We randomly selected hundered DLPMs i.e. $\mathbf{a}^*$'s. We then used Algorithm 2 with $\epsilon = 0.01$ to recover the estimates $\hat{\mathbf{a}}$'s. In Table 7, we report the proportion of the number of times $\|\mathbf{a}^* - \hat{\mathbf{a}}\|_\infty \leq \omega$ for different values of $\omega$. We see improved elicitation when we work with $\lambda$ and incur more errors when the sphere's radius is less than that. In particular, if we take the radius of the order (a little) higher than $10^{-12}$ then we perform perfect elicitation. Needless to say, when working with real oracle (users), the magnitude of the oracle's feedback noise $\epsilon_\Omega$ and the size of the sphere will play a role in elicitation performance as suggested in Theorem 2.

Table 7: LPM elicitation on sphere with varying (small) radius and $\epsilon = 0.01$. For randomly chosen hundred $\mathbf{a}^*$, we show the fraction of times our estimates $\hat{\mathbf{a}}$ obtained with $4 \times 2(q-1)\lceil \log(1/\epsilon) \rceil$ queries satisfy $\|\mathbf{a}^* - \hat{\mathbf{a}}\|_\infty \leq \omega$. Notice that we incur error only when the radius is of the order of practical computation error, which can be attributed to $\epsilon_\Omega$ in the simulated setting.

| $\lambda$ \ $\omega$ | 0.02 | 0.04 | 0.06 | 0.08 | 0.10 |
|---|---|---|---|---|---|
| $1.250 \times 10^{-12}$ | 0.03 | 0.38 | 0.74 | 0.92 | 0.94 |
| $1.875 \times 10^{-12}$ | 0.09 | 0.49 | 0.77 | 0.94 | 0.98 |
| $2.500 \times 10^{-12}$ | 0.12 | 0.73 | 0.93 | 0.97 | 0.99 |

Table 8: DLFPM Elicitation for synthetic distribution for $k = 3$ classes (Appendix F.3) with $\epsilon = 0.01$. $(\mathbf{a}^*, \mathbf{b}^*, b_0^*)$ denote the true DLFPM. $(\hat{\mathbf{a}}, \hat{\mathbf{b}}, \hat{b}_0)$ denote the elicited LFPM. $\alpha$ and $\sigma$ denote the mean and the standard deviation in the ratio of the elicited to the true metric (evaluated on the confusions in $\partial\mathcal{D}$ used in Algorithm 5), respectively. We empirically verify that the elicited metric is constant multiple ($\alpha$) of the true metric.

| True Metric | Results on Synthetic Distribution (Appendix F.3) | | |
|---|---|---|---|
| $(a_1^*, a_2^*, a_3^*), (b_1^*, b_2^*, b_3^*), b_0^*$ | $(\hat{a}_1, \hat{a}_2, \hat{a}_3), (\hat{b}_1, \hat{b}_2, \hat{b}_3), \hat{b}_0$ | $\alpha$ | $\sigma$ |
| (0.21, 0.59, 0.20), (0.11, -0.22, -0.27), 0.41 | (0.25, 0.58, 0.18), (0.20, -0.03, -0.17), 0.29 | 1.23 | 0.03 |
| (0.45, 0.27, 0.29), (0.39, 0.22, -0.76), 0.43 | (0.46, 0.34, 0.20), (0.42, 0.30, -0.73), 0.38 | 1.03 | 0.04 |
| (0.08, 0.42, 0.50), (0.07, -0.63, 0.20), 0.37 | (0.16, 0.38, 0.47), (0.17, -0.41, 0.23), 0.27 | 1.22 | 0.05 |

## F.3   DLFPM and LFPM Elicitation

Now, we validate elicitation for DLFPMs for classes $k = 3$ and $k = 4$ using the routine discussed in Appendix E.1. We use the same distribution setting of Section 6.1 for both the classes. We define a true metric $\psi^*$ by $\{\mathbf{a}^*, \mathbf{b}^*, b_0^*\}$. Then, we run Algorithm 1 with $\epsilon = 0.01$ to find the hyperplane $\bar{\ell}$ and maximizer on $\partial\mathcal{D}^+$, Algorithm 4 with $\epsilon = 0.01$ to find the hyperplane $\underline{\ell}$ and minimizer on $\partial\mathcal{D}^-$, and Algorithm 5 with $n' = 1000$ (1000 diagonal confusions on $\partial\mathcal{D}^+$ obtained by varying parameter $m$) and $\delta = 0.01$. This gives us the elicited metric $\hat{\psi}$, which we represent by $\{\hat{\mathbf{a}}, \hat{\mathbf{b}}, \hat{b}_0\}$. In Table 8 and Table 9, we present the elicitation results for DLFPMs for classes $k = 3$ and $k = 4$, respectively. We also present the mean ($\alpha$) and the standard deviation ($\sigma$) of the ratio of the elicited metric $\hat{\psi}$ to the true metric $\psi^*$ over the set of diagonal confusions used in Algorithm 5(column 3 and 4 of Table 8 and Table 9). For a better judgment, we show function evaluations of the true metric and the elicited metric in Figure 7. The true and the elicited metric are plotted together after vectorizing the set of diagonal confusions in a certain order based on their parametrizations. As expected, we see that the elicited metric is a constant multiple of the true metric.

Now, we validate elicitation for LFPMs for classes $k = 3$ and $k = 4$ using the routine discussed in Appendix E.2. We define a true metric $\phi^*$ by $\{\mathbf{a}^*, \mathbf{b}^*, b_0^*\}$. Then, we run Algorithm 2 with $\epsilon = 0.01$ to find the hyperplane $\bar{\ell}$ and maximizer on $\partial\mathcal{S}_\lambda^-$, Algorithm 6 with $\epsilon = 0.01$ to find the hyperplane $\underline{\ell}$ and minimizer on $\partial\mathcal{S}_\lambda^+$, and Algorithm 7 with $n' = 1000$ (1000 off-diagonal confusions on $\partial\mathcal{S}_\lambda^-$ obtained by varying parameter $\boldsymbol{\theta}$) and $\delta = 0.01$. This gives us the elicited metric $\hat{\phi}$, which we represent by $\{\hat{\mathbf{a}}, \hat{\mathbf{b}}, \hat{b}_0\}$. In Table 10, we present the elicitation results for LFPMs for classes $k = 3$. We also present the mean ($\alpha$) and the standard deviation ($\sigma$) of the ratio of the elicited metric $\hat{\phi}$ to the true metric $\phi^*$ over the set of off-diagonal confusions used in Algorithm 7 (column 3 and 4 of Table 10). For a better judgment, we show function evaluations of the true metric and the elicited metric evaluated on selected off-diagonal confusions in the top row of Figure 8. Due to many terms in the LFPM for $k = 4$, we skip providing true metric and the elicited metric and only mention the $\alpha$ and $\sigma$ of the true and elicited metric similar to Table 10. We obtained $\alpha = 0.79, 0.72, 0.72$ and $\sigma = 0.007, 0.007, 0.006$ for the three metrics plotted in the bottom row of Figure 8. The true and the elicited metric are plotted together after vectorizing the set of confusions in a certain order based on their parametrizations. As expected, we again see that the elicited metric is a constant multiple of the true metric for both $k = 3$ and $k = 4$.

Table 9: DLFPM Elicitation for synthetic distribution for $k = 4$ classes (Appendix F.3) with $\epsilon = 0.01$. $(\mathbf{a}^*, \mathbf{b}^*, b_0^*)$ denote the true DLFPM. $(\hat{\mathbf{a}}, \hat{\mathbf{b}}, \hat{b}_0)$ denote the elicited LFPM. $\alpha$ and $\sigma$ denote the mean and the standard deviation in the ratio of the elicited to the true metric (evaluated on the diagonal confusions in $\partial\mathcal{D}$ used in Algorithm 5), respectively. We empirically verify that the elicited metric is constant multiple ($\alpha$) of the true metric.

| True Metric | Results on Synthetic Distribution (Appendix F.3) | | |
|---|---|---|---|
| $(a_1^*, a_2^*, a_3^*, a_4^*)$, $(b_1^*, b_2^*, b_3^*, b_4^*), b_0^*$ | $(\hat{a}_1, \hat{a}_2, \hat{a}_3, \hat{a}_4)$, $(\hat{b}_1, \hat{b}_2, \hat{b}_3, \hat{b}_4), \hat{b}_0$ | $\alpha$ | $\sigma$ |
| (0.32, 0.35, 0.06, 0.27), (-1, -0.3, -0.32, 0.25), 0.6 | (0.2, 0.29, 0.19, 0.32), (-0.4, -0.01, 0.08, 0.33), 0.26 | 1.58 | 0.12 |
| (0.31, 0.22, 0.27, 0.2), (-0.17, -0.01, 0.18, 0.09), 0.25 | (0.2, 0.3, 0.26, 0.24), (-0.38, 0.07, 0.16, 0.14), 0.28 | 0.95 | 0.04 |
| (0.22, 0.16, 0.41, 0.21), (-0.22, -0.43, -0.18, 0.14), 0.33 | (0.19, 0.2, 0.35, 0.26), (-0.09, -0.12, -0.03, 0.24), 0.19 | 1.38 | 0.06 |

Table 10: LFPM Elicitation (Appendix F.3) for $k = 3$ classes with $\epsilon = 0.01$. $(\mathbf{a}^*, \mathbf{b}^*, b_0^*)$ denote the true LFPM. Notice that there are thirteen terms to elicit in LFPM. $(\hat{\mathbf{a}}, \hat{\mathbf{b}}, \hat{b}_0)$ denote the elicited LFPM. $\alpha$ and $\sigma$ denote the mean and the standard deviation in the ratio of the elicited to the true metric (evaluated on the diagonal confusions in $\partial\mathcal{D}$ used in Algorithm 7), respectively. We empirically verify that the elicited metric is constant multiple ($\alpha$) of the true metric.

| True Metric | Results on Synthetic Distribution (Appendix F.3) | | |
|---|---|---|---|
| $(a_1^*, a_2^*, a_3^*, a_4^*, a_5^*, a_6^*)$, $(b_1^*, b_2^*, b_3^*, b_4^*, b_5^*, b_6^*), b_0^*$ | $(\hat{a}_1, \hat{a}_2, \hat{a}_3, \hat{a}_4, \hat{a}_5, \hat{a}_6)$, $(\hat{b}_1, \hat{b}_2, \hat{b}_3, \hat{b}_4, \hat{b}_5, \hat{b}_6), \hat{b}_0$ | $\alpha$ | $\sigma$ |
| (-0.16, -0.05, -0.29, -0.21, -0.17, -0.12), (-0.76, 0.02, -0.88, 0.09, -0.23, -0.38), 2.36 | (-0.11, -0.08, -0.15, -0.17, -0.24, -0.25), (-0.66, 0.07, -0.86, 0.04, -0.04, -0.09), 1.89 | 1.11 | 0.01 |
| (-0.17, -0.19, -0.09, -0.18, -0.16, -0.2), (-0.3, -0.74, -0.54, -0.37, -0.89, -0.14), 2.99 | (-0.05, -0.08, -0.11, -0.16, -0.31, -0.31), (-0.46, -0.82, -0.43, -0.34, -0.48, 0.09), 2.58 | 1.08 | 0.01 |
| (-0.3, -0.08, -0.1, -0.12, -0.21, -0.18), (-0.24, -0.52, -0.45, 0, -0.41, -0.94), 2.67 | (-0.06, -0.08, -0.11, -0.15, -0.27, -0.33), (-0.59, -0.45, -0.37, 0.07, -0.24, -0.57), 2.36 | 1.07 | 0.01 |

(a) Table 8, Line 1     (b) Table 8, Line 2     (c) Table 8, Line 3

(d) Table 9, Line 1     (e) Table 9, Line 2     (f) Table 9, Line 3

Figure 7: True and elicited DLFPMs for synthetic distribution from Table 8. The solid green curve and the dashed blue curve are the true and the elicited metric, respectively. We plot the metrics on a vectorized set of diagonal confusions i.e. confusion matrices are sorted by their parametrizations ($m$) in a particular way. We see that the elicited DLFPMs are constant multiple of the true metrics for both $k = 3$ and $k = 4$.

| | | |
|---|---|---|
| (a) Table 10, Line 1 | (b) Table 10, Line 2 | (c) Table 10, Line 3 |
| (d) LFP Metric 1, $k = 4$ | (e) LFP Metric 2, $k = 4$ | (f) LFP Metric 3, $k = 4$ |

Figure 8: True and elicited LFPMs. The plots in the top row correspond to the metrics in Table 3 for $k = 3$. The bottom row corresponds to metrics for $k = 4$ (due to many terms, we only provide the plots for $k = 4$). The solid green curve and the dashed blue curve are the true and the elicited metric, respectively. We plot the metrics on a vectorized set of off-diagonal confusions i.e. confusions are sorted by their parametrizations ($\boldsymbol{\theta}$) in a particular way. We see that the elicited LFPMs are constant multiple of the true metrics for both $k = 3$ and $k = 4$.

## Footnotes

[2]The datasets can be downloaded from: www.csie.ntu.edu.tw/ cjlin/libsvmtools/datasets/multiclass.html