[Reviews · NeurIPS 2019]

Reviewer 1



This paper studies the metric elicitation problem, a problem proposed by [7] (AISTATS'19 paper). Unlike the previous work on the binary classification, this paper generalizes to the multi-class setting. The authors defined two types of metrics for multi-class classification, Diagonal Linear Performance Metric (DLPM) and Linear Performance Metric (LPM), and designed algorithms for the two type of metrics respectively. The algorithm for DLPM is relatively simple: just use binary-search like [7] and apply it for (k-1) times. The LPM case is much more complicated - the authors proposed a coordinate-wise binary-search style algorithm based on the geometry of the feasible set. Theoretical guarantees for both algorithms are provided. About the presentation of the main results: I am a little confused by the presentation of the main results. To me, it seems like the most interesting (and practical) performance metric, is the linear-fractional one in Appendix E.2. Why isn't this (perhaps the most important one) presented and highlighted in the main text? Is there any particular reason for it? Disclaimer: I have NOT checked the (very long!) appendix very carefully. About the assumptions: the assumptions in the main text (assumptions 1&2) look standard to me. But in the appendix, I am not sure why assumptions 3&4 are necessary - they seem quite unnatural to me. Since these linear-fractional settings are the more important ones, I hope the authors can explain a bit on this. About the optimality of the bounds: It is nice that the proposed algorithms all have theoretical guarantees. But how good are these algorithms? In [7], there is a simple O(log(1/eps)) lower bound on query complexity for binary classification. However, in this paper, I cannot find any lower bounds. It does seem like the O(k) and O(log(1/eps)) is necessary in theorem 1 (but I still hope that we can have a formal lower bound). But for the dependency on the feedback noise eps_omega, theorem 1 has a \sqrt(eps_omega) overhead on the error, but theorem 2 has a sqrt(q*eps_omega) = O(k* \sqrt(eps_omega)). Note that this term is important since it does not go to zero when eps goes to zero. How necessary is the extra factor of k here? Overall, with the concerns above, I think this paper is marginally below the acceptance threshold. But I am happy to increase my score if the authors can address these issues in the response. ======================== After the author response, I think the authors addressed part of my concerns, so I raised my score by 1pt. However, I agree with reviewer 2 that the writing and experiments can be improved, so I won't increase my score beyond that.

Reviewer 2



The authors consider the setting of [1'] where the goal is to find the implicit performance measure that an expert uses for classification. This performance metric is assumed to be a linear function of the confusion matrix (or occasionally a linear fractional as in [1]) and the goal is to find it (or approximate it) using a few queries. The queries are in the form of comparison queries which given two classifiers (or confusion matrices) outputs the classifier that is better according to the implicit score function. This submission extends the analysis of [1] to the multiclass case which poses new challenges. The algorithm for the diagonal case finds the ratios between one of the diagonal elements (say the first one) and all the other elements. This is sufficient as metric is scale invariant. Finding each of the ratios is a simple binary search problem (though the analysis requires some considerations regarding the geometry of the problem to make sure that the binary search method works). The algorithm for the general (linear) case is based on binary search and the ideas from the derivative-free optimization approach. Some experiments have been done. However, no comparisons with any baselines have been made. The authors can at least compare the method with a method that queries randomly and picks the winner (and again the winner is compared with another random solution). I think there are many other better baselines to compare with. The work of [2] may also be somewhat relevant. [1'] G. Hiranandani, S. Boodaghians, R. Mehta, and O. Koyejo. Performance metric elicitation from pairwise classifier comparisons. In The 22nd International Conference on Artificial Intelligence and Statistics, pages 371–379, 2019. [2] Kane et al, "Active classification with comparison queries" === The author's response did not change my decision. I still think the experimental section is weak and the paper needs some rewrite.

Reviewer 3



# originality Even though metric elicitation itself has been proposed by Hiranandani et al. (2019) in the context of binary classification, there is still a large gap between binary and multiclass classification. I claim that there still exists novelty in terms of multiclass analysis of metric elicitation such as restricted Bayes optimal classifiers. # quality Although I could not check every detail of proofs, the proposed algorithms seem reasonable to me. On the other hand, I want to confirm one point. Why do you query four points in both algorithms 1 and 2 although those algorithms are based on binary search? I have read Appendix A and found that you mentioned this is because it would successfully escape from flat regions. However, I guess we can still construct pathological cases that do not work for algorithms querying four points. For example, assume that we query four times and find that m^a = m^c = m^d = m^e = m^b. Even so, there might be a peak in between, say, m^a and m^c. I guess that phenomena would be difficult to avoid no matter how many query times we increase, as long as the target metric parameterization is pseudo concave. Could you clarify this point? # clarity - Definitions 1 and 2: I still did not get the intuition on why the norms are different. Is it due to the different parameterization of confusion matrices? - In "Parameterization of upper boundary" (l.149): I would prefer adding \nu: [0,1] \to \partial\mathcal{D}^+_{k_1,k_2}. The same to the parameterization in LPM (l.183). - l.242: It would be better to add "1-Lipschitz with respect to the confusion matrices." In addition, why does this hold with high probability, not with probability 1? # significance Multiclass metric elicitation would have a practical impact because it gives one way to choose an appropriate performance metric, which is intuitively quite hard. The extension technique to the multiclass case may help further extensions of metric elicitation to more complicated tasks, such as multi-label and structured predictions. ========== After feedback ========== Thank you for responding to my questions. On the first answer, claiming the unimodality has great importance. In addition, it is interesting to look at Figure 10. However, I still have a concern that we cannot detect the modal no matter how many queries we use, given the illustration in Figure 10. For example, take a look at Figure 4 (right) in Appendix. If we obtained this kind of oracle responses, we are still not aware that the modal is located in between m^a and m^c, or between m^c and m^d. Anyway, since I love the idea presented in the current paper, I hope this part can be made clearer in the future version. Here, I will not change the score.

[Author Response · NeurIPS 2019]

We thank the reviewers for careful examination of our paper. Since there are no common concerns, we address individual concerns. For the rebuttal, we use references from the main paper plus some references added here.

**Reviewer 1.** **1. LFPM Elicitation and Significant Contributions:** In our experience, linear metrics are by far the most used in practice (see [A, 21] and references therein), so we chose to focus on this case. Even for the linear case, there are many subtle issues that we address – including a novel characterization of the space of confusion matrices, introducing and analysing restricted Bayes optimal classifiers, developing algorithms with theoretical guarantees, and showing robustness for the practical applications (noise analysis). We hope the reviewer agrees, in accord with the other reviewers, that these are significant contributions. We also note that the linear case is important for understanding more complex settings, though all the additional details are difficult to compress into eight pages. However, we agree with the reviewer that LFPM elicitation is still important, and therefore, instead of discarding it completely, we summarized it in Section 7 and discussed it thoroughly in the appendix to conclude our scientific contributions.

**2. Assumption 3 and 4:** These are sufficient conditions for DLFPMs (LFPMs) to be bounded and monotonically increasing (decreasing) in diagonal (off-diagonal) elements of the confusion matrices. This is detailed in proof of Proposition 5 (Proposition 7). It is equivalent to fixing $\|\mathbf{a}\|_1 = 1$, $a_i \geq 0$ for the diagonal linear case (Section 2.2). The only additional restriction for the linear-fractional case is $b_0 = \sum_i (a_i - b_i)\zeta_i$, instead of the derived condition $b_0 \geq \sum_i (a_i - b_i)\zeta_i$ (see line 614), which is sufficient to guarantee a unique metric bounded in $[0, 1]$ (instead of one of the equivalent alternatives). Note that most existing linear-fractional metrics satisfy these conditions [7, 11, 12].

**3. Lower Bound:** We conjecture that our bounds are tight (section 7), but we leave a proof for future work. Our initial analysis says that it requires an additional understanding of the query space. We hope the reviewer agrees that query complexity bounds are important even when lower bounds are yet unknown.

**4. Factor of k:** Notice that the error guarantee in Theorem 1 is in $\|\cdot\|_\infty$-norm; whereas, it is in $\|\cdot\|_2$-norm in Theorem 2. Thus, using standard norm bounds, it is clear that both have square root dependence on the number of unknown terms in $\|\cdot\|_2$-norm. We thank the reviewer for pointing this out and will clarify in the final version.

Figure 9: Passive Learning.

**Reviewer 2.** **1. Experiments:** Our experiments are primarily designed to empirically validate our theory. Since this is the first work on multiclass ME, we are unaware of any baselines. The suggested strategy of posing random queries is easily shown to require exponential time to achieve $\epsilon$ error (using $\epsilon$-ball finite parcellation of the space of confusion matrices), thus is extremely query-inefficient. In Section 8, we outline several approaches which learn linear functions from pairwise comparisons in a passive manner [9, 6, 14] i.e. by first randomly collecting pairwise comparisons and then learning a linear function $\hat{\mathbf{a}}^T \mathbf{c}$. To verify the inferiority of the passive approach, we present the performance of [9] for the two metrics (for $k = 3, 4$) in row 1 of Table 2, and plot the error $\|\mathbf{a}^* - \hat{\mathbf{a}}\|_\infty$ in Figure 9. The plot is averaged over 5 random runs. We see that even after 400 queries the error is greater than 0.1 for the baseline; whereas, we only require 56 (resp. 84) queries for $k = 3$ (resp. $k = 4$) to achieve 0.01 error. While we chose not to compare to these trivial baselines, if the reviewer strongly feels these experiments are helpful for a broad audience, we are happy to add such experiments in the additional page of the final version.

**2. Relevant Paper [B]:** Comparison queries in [B] solve a different problem of actively finding a good classifier (wrt. the accuracy metric), compared to our problem of finding the oracle's metric. However, we believe some ideas from [B] may be relevant, and we would like to thank the reviewer for the reference. We will add it in the final version.

Figure 10: Two Queries

**Reviewer 4.** **1. Four Queries in Algorithm 1:** Unlike the standard binary search, we want to find the mode of a unimodal function using pairwise comparisons. Note that posing two queries in each iteration does not achieve the goal. As an example, compare the solid and dotted functions in Figure 10. Since the query responses will be same for both functions, we cannot decide the next search interval. Thus, we need more than two queries. On the other hand, we would like to thank the reviewer for pointing our unclear description of *unimodal*. Notice that due to Assumption 1, every supporting hyperplane of $\mathcal{D}_{k_1,k_2}$ supports a unique point on the boundary $\partial \mathcal{D}_{k_1,k_2}^+$ and vice-versa (Proposition 1); therefore, we indeed do not have flat regions. We will clarify this in the final version.

**2. Difference in norms:** The norms were chosen to best complement the underlying metric elicitation algorithm and vice-versa. For example, wlog, we can assume $\|\cdot\|_2$ normalization in Definition 1, but then the form of the solution becomes a little complex. If desired, we are happy to transform results to various norms using standard norm bounds.

**3. $\nu, \mu$ details:** Thank you for the suggestion. We will add these details in the final version.

**4. With high probability argument:** When working with finite samples, we cannot guarantee that the estimate of confusion matrix $\hat{\mathbf{c}}$ will converge to the true $\mathbf{c}$ with probability 1 due to finite sample effects. Now notice that since the oracle response $\Omega(\hat{\mathbf{c}}, \hat{\mathbf{c}}') = \mathbb{1}[\phi(\hat{\mathbf{c}}) > \phi(\hat{\mathbf{c}}')]$ is a 1-Lipschitz function of the confusion matrices, we can guarantee correct feedback i.e. $\Omega(\mathbf{c}, \mathbf{c}') = \mathbb{1}[\phi(\mathbf{c}) > \phi(\mathbf{c}')]$ only with high probability (not with probability 1).

[A] Elkan, Charles. "The foundations of cost-sensitive learning." IJCAI, 2001.

[B] Kane, Daniel M., et al. "Active classification with comparison queries." FOCS, 2017.


[Meta-Review · NeurIPS 2019]

The reviewers agree that the paper's theoretical contribution is adequate, although it may require some rewrite as well as more detailed experiments. It may be useful to elaborate results on the linear-fractional performance metric, as these are nontrivial and relevant in practice.